# Therapeutic Strategies to Activate p53

**DOI:** 10.3390/ph16010024

**Published:** 2022-12-24

**Authors:** Angelo Aguilar, Shaomeng Wang

**Affiliations:** The Rogel Cancer Center, Departments of Internal Medicine, Pharmacology and Medicinal Chemistry, University of Michigan, Ann Arbor, MI 48109, USA

**Keywords:** wild-type p53, mutant p53, activators, MDM2, MDMX, Y220C, inhibitors, PROTACS, degraders, molecular glue

## Abstract

The p53 protein has appropriately been named the “guardian of the genome”. In almost all human cancers, the powerful tumor suppressor function of p53 is compromised by a variety of mechanisms, including mutations with either loss of function or gain of function and inhibition by its negative regulators MDM2 and/or MDMX. We review herein the progress made on different therapeutic strategies for targeting p53.

## 1. Introduction

Appropriately named the guardian of the genome, p53 is the hub in which cell fate is determined [1]. p53 is a transcription factor that functions as a tumor suppressor, which is induced in response to various stresses, such as DNA damage, oncogene expression, oxidative stress, nutrient deprivation, hypoxia, telomere attrition, or ribosomal dysfunction through both transcription-dependent and -independent mechanisms [2]. In response to these stresses, p53 can activate or repress the expression of many target genes that induce responses, such as cell cycle arrest, DNA repair, senescence, apoptosis, metabolism, autophagy, or ferroptosis [3].

The p53 protein has 393 amino acids, which are encoded by the *TP53* gene. Structurally, p53 comprises several domains with different functions, including: an N-terminal transactivation domain (1–62) and proline rich region (63–94); a DNA binding domain (95–292); an oligomerization domain (325–356); and a C-terminal regulatory domain (357–393) (Figure 1) [2,4,5]. In unstressed cells, low levels of p53 are maintained by direct interactions with its negative regulators MDM2 and MDMX [6,7,8]. However, upon stress stimuli, such as DNA damage or oncogene activation, p53 is phosphorylated (P) by upstream kinases [9,10] or acetylated (Ac) [11,12]. Both of these post-translational modifications (PTM) stabilize and activate p53 by dislocating it from and decreasing the binding to its negative regulator MDM2 [9,10,11,12]. p53 is highly disordered and consequently has intrinsically low thermodynamic and kinetic stability and a body temperature half-life of about 9 min during which it switches rapidly between its unfolded and folded states [13,14]. In the activation pathway of p53, heat shock proteins (HSP) function as molecular chaperones [15,16,17] that assist p53 in folding and adopting its active, tetrameric form [18,19,20,21,22], which binds to targeted DNAs for gene transcription (Figure 1). 

In essentially almost all human cancers, the powerful tumor suppressor function of p53 is compromised through a variety of mechanisms, such as amplification or upregulation of its negative regulators MDM2 and/or MDMX, or through the mutation or deletion of the *TP53* gene. In the last 20 years, significant efforts have been made to reactivate the tumor suppressor function of both wild-type (wt) and mutant p53.

The therapeutic strategies targeting p53 can be broadly divided into two categories: (i) activation of wt-p53 and (ii) restoration of wt-like functions of mutant p53, which have been the subjects of a number of reviews [2,3,23,24,25,26]. In this review, we will focus on recent therapeutic advances in targeting the major negative regulation MDM2–MDMX–p53 axis and activation of one specific p53 Y220C mutant.

## 2. Targeting the Protein–Protein Interaction between MDM2 and p53

In approximately 50% of human cancers, p53 retains its wild-type (wt) status, but its functions are effectively inhibited by the murine double minute 2 (MDM2) protein through a direct protein–protein interaction [27,28,29,30,31,32,33]. Through direct binding, the MDM2 protein blocks the transactivation domain of p53, mono-ubiquitinates it to transport p53 from the nucleus to the cytoplasm, and then poly-ubiquitinates p53 for proteasomal degradation [27]. The interaction of MDM2 and p53 occurs between the first 120 N-terminal amino acids of MDM2 and the first 30 N-terminal amino acids in the transactivation domain of p53. A high-resolution co-crystal structure of residues 15–29 of p53 bound to MDM2 has been reported [31,34]. In this co-crystal structure, the p53 peptide adopts an α-helical conformation, which brings its three hydrophobic residues Phe19, Trp23, and Leu26 together to form a cluster, which interacts with a well-defined, hydrophobic pocket in MDM2. The discovery of this well-defined pocket in MDM2 suggested that the MDM2–p53 interaction may be effectively targeted by small-molecule inhibitors, which has been the subject of extensive research efforts by the academia and industry in the last 20 years. To date, a number of potent and selective MDM2 inhibitors have been progressed into clinical development (Figure 2). Here, we summarize a number of successful programs that led to the advancement of MDM2 inhibitors into clinical trials.

Among the first in class MDM2 inhibitors that were reported are the Nutlins, which were discovered by Hoffmann-La Roche through screening of a library of synthetic chemicals. Further chemical optimization to improve potency and selectivity produced the first lead compound, Nutlin-3a [35]. Further optimization led to the discovery of RG-7112 (Figure 2 and Figure 3A), which has an MDM2 binding affinity with IC_50_ = 18 nM and an average cellular activity with IC_50_ = 400 nM in wt-p53 cancer cells (Table 1). RG-7112 was the first MDM2 inhibitor to enter clinical trials [36]. In the reported optimization SAR, substitution of the 4 and 5 positions of the imidazoline core of Nutlin-3a with methyl groups was accomplished to prevent oxidation, which converts this core of the compound into an inactive imidazole. The *tert*-butyl group in RG7112 was introduced to replace the methoxyl group, which was found to be a major metabolic soft spot, producing the phenol metabolite. The isopropyl group in Nutlin-3a that enters the Phe19 pocket was replaced with an ethyl group to decrease molecular weight, while the piperazine substituent was modified to further improve the binding and pharmacokinetic properties, leading to the optimized RG-7112 that entered clinical trials [36] (Figure 3A). Its clinical trial demonstrated that MDM2 inhibitors can indeed activate p53 signaling in human tumors, validating this approach.

The Wang lab at the University of Michigan, inspired by structures of natural products and rational design, discovered the spiro-oxindoles as a novel class of MDM2 inhibitors, with MI-219 as the initial lead [50,51,52]. Based on the chemistry used to build the spiro-oxindole scaffold, the stereochemistry around the pyrrolidine ring of MI-219 was assigned, but subsequently, it was discovered that a rapid and reversible retro-Mannich and recyclization reaction converted the spiro-oxindoles to their more potent and energetically stable all-*trans* isomers (Figure 3B) [53,54]. Further optimization of this class of compounds led to the discovery of MI-773, which was entered into clinical trials [37]. The optimization of this compound was mainly achieved by extensive modification of the carboxamide substituent, which improved its potency, pharmacokinetic properties, and in vivo efficacy. Its successful co-crystallization with MDM2 revealed the reason for its superior potency, showing that MI-773 establishes additional interactions with the MDM2 protein, which are not experienced by the Nutlins (Figure 3B). In addition to occupying the Phe19, Trp23, Leu26 pockets in MDM2 through the substituents on the core pyrrolidine of MI-773, the N-H of the oxindole forms a hydrogen bond with the backbone carbonyl of Leu54; the carbonyl of the carboxamide functions as a hydrogen bond acceptor from the NH of Hys96, and the 4-hydroxyl-cyclohexyl-carboxamide substituent has additional hydrophobic interaction with the MDM2 surface, and its hydroxyl group forms a hydrogen bond with Lys94 of the MDM2 protein (Figure 3B). Capturing these additional hydrophobic and polar interactions together with the three key p53 interactions contributes to this class of compounds having a significantly enhanced MDM2 binding affinity (K_i_ = 0.88 nM) and cellular activity (IC_50_ < 200 nM) in wt-p53 cancer cells (Table 1). This compound showed good pharmacokinetics and achieved complete tumor regression in mouse xenograft models with no signs of toxicity [37].

After discovering the Nutlins, the Hoffmann-La Roche team continued their exploration of novel MDM2 inhibitors, including the investigation of spiro-oxindole analogs [55]. Inspired by the highly functionalized pyrrolidine core present in the spiro-oxindole class of MDM2 inhibitors and through de novo design around the pyrrolidine ring, they explored different configurations of the pyrrolidine ring substituents and found that the trans-configuration between the aryl groups gave the best results [38]. It was found that the CN group was critical to achieving this trans-configuration, and this led to the discovery of a new series of MDM2 inhibitors with a novel pyrrolidine core. This exploration yielded RG7388 (Figure 3C), which has an MDM2 binding affinity (IC_50_ = 6 nM) and an average cellular activity (IC_50_ = 30 nM) in wt-p53 cancer cells (Table 1). RG7388 is now in clinical trials [38]. The co-crystal structure of RG7388 with MDM2 was not obtained, but that of a close analog shows its likely binding mode (Figure 3C). RG7388 has almost the same substituents as MI-773, which occupy the Phe19, Trp23, and Leu26 pockets, with the exception of the spiro-oxindole portion, which is replaced and mimicked by the quaternary carbon-3, which is substituted with a cyano and a 2-fluoro-4-chlorophenyl group. Furthermore, in RG7388, the carboxamide substituent was switched from an aliphatic group to a *meta*-methoxylbenzoic acid. Not surprisingly, the binding mode of the RG7388 class to MDM2 was identical to that of MI-773, with the exception that RG7388 does not possess the oxindole, which can form a hydrogen bond with the carbonyl of Leu54 of the MDM2 protein. It was reported, however, that replacement of the aliphatic with an aromatic carboxamide substituent was critical to metabolic stability and, in this class of molecules, significantly improved the cellular potency, PK profile, and in vivo efficacy [38]. The loss of the oxindole hydrogen bonding with Leu54 in RG7388 is probably compensated for by the increased potency from the 3-methoxy substituent on the aromatic carboxamide and the strong electrostatic interaction of the carboxylate with Lys94 of MDM2, which is only a hydrogen bond in the co-crystal structure involving MI-773. RG7388 is the MDM2 inhibitor, which has progressed the furthest and is currently in phase III clinical trials. 

Inspired by available structural information, Amgen scientists used a de novo design approach in their discovery of the trans-piperidone class of MDM2 inhibitors, which produced the clinical compound AMG232 (Figure 4A) [39,40]. Reports from this group revealed the power of rational design using available structural information for the initial design, then further optimization guided by a combination of modeling and co-crystal structures of compounds in the series. As a consequence, each substituent in the structure has a specific interaction with the MDM2 protein and helps maintain adequate physicochemical properties for a drug candidate. AMG232 has an MDM2 binding affinity (IC_50_ = 0.6 nM) and an EdU cell proliferation (IC_50_ = 9.1 nM) in SJSA-1 cancer cells (Table 1). The co-crystal structure of AMG232 bound to MDM2 has not been reported, but that of a close analog shows the key interactions (Figure 4A) [39]. This co-crystal structure indicates that the isopropyl, *para*-chlorophenyl, and *meta*-chlorophenyl substituents occupy the three key binding pockets on MDM2, mimicking the interactions of the p53 residues Phe19, Trp23, and Leu26, respectively. Its acetic acid functionality captures a strong electrostatic interaction with the imidazole of His96 of MDM2, and this significantly contributes to potency. As explained in their report, a methyl substituent was incorporated at C3 to introduce a 1,3-steric strain and destabilize the undesired anti-like conformation between the two aryl groups, leading to a dominant *gauche* conformation, which resulted in a further increase in the potency [40]. Unique to this class of compounds is their capture of hydrophobic interactions in the Gly58 shelf with the isopropyl group associated with the sulfone moiety. They explained that the incorporation of this sulfone moiety served two roles. The first role is that this hydrophilic group stabilizes the conformation required to direct the N-alkyl substituent into the Phe19(p53) binding pocket and the second is to capture the interaction in the Gly58 shelf of the MDM2 protein to further improve potency and maintain good metabolic stability [40]. 

In their pursuit of novel small-molecule MDM2 inhibitors, a Novartis group used a design strategy, which revolves around the “central valine concept”. This concept involves placing a tailored aromatic core close to the Val93 residue in the center of the p53 binding pocket in MDM2 and then substituting positions around the core with appropriate vectors toward the Phe19, Trp23, and Leu26 binding pockets in MDM2 [56]. Virtual screening of their internal collection of compounds resulted in a preliminary selection of around 50,000 compounds, and from these, a dihydroisoquinolinone hit was identified. Through medicinal chemistry exploration, they discovered a new class of dihydroisoquinolinone derivatives as novel MDM2 inhibitors [57]. The major optimization efforts with this class of inhibitors focused on physicochemical properties, which influence in vivo toxicological outcomes, and yielded NVP-CGM097 with an MDM2 binding affinity of IC_50_ = 1.7 nM and cell activity of IC_50_ < 450 nM in wt-p53 cancer cells (Table 1). NVP-CGM097 was advanced to clinical trials [41]. Interestingly, the co-crystal structure of NVP-CGM097 revealed an unprecedented binding mode, which is the most distinct binding mode of all the MDM2 inhibitors. Most of its core lies close to and makes hydrophobic contact with residues Ile54, Phe55, and Gly58 of the MDM2 protein. This is the surface wall within the binding cleft and is opposite to that occupied by most of the other MDM2 inhibitors, which interact with the wall containing Val93 (Figure 4B). Even with this unusual binding mode, the substituents of NVP-CGM097 still capture the three key interactions that are necessary for binding to MDM2. The unique features of this scaffold are that it causes the Phe-55 residue of MDM2 to swing up to form a face-to-edge interaction with the dihydroisoquinolinone core and also that the N-methyl piperazinone substituent is nestled between the MDM2 protein walls above the Phe19 binding pocket; these two interactions were reported to contribute significantly to the potency of NVP-CGM097. Interestingly, it was also reported that due to its distinct binding mode to MDM2, a species-dependent binding was observed, which was not seen with Nutlins and presumably not with other MDM2 inhibitors whose binding is similar to that of Nutlins [41]. This difference was attributed to two amino acid residues, Leu54 and Leu57, in the human sequence, which are both switched to isoleucines in rodents, and in the case of dogs, only Leu54 is switched to isoleucine. This additional methyl on isoleucine causes a steric clash with the inhibitor, decreasing its binding in rodents and dogs but retaining its potency in humans and monkeys [41]. NVP-CGM097 was the first MDM2 inhibitor advanced into clinical trials by Novartis.

Continuing with their efforts to discover novel inhibitors of MDM2, the Novartis scientists proceeded to re-engineer the dihydroisoquinolinone mainly because its binding mode unexpectedly did not fit the “central valine” binding mode, making it an inefficient binder to MDM2. Deviation from this binding mode was rationalized as the result of a bent core, which results in a conformation energy loss in its binding to MDM2, which is compensated by other van der Waals and hydrogen bonding interactions with the MDM2 binding site [42,58]. This re-engineering produced the imidazolopyrrolidinone analog NVP-HDM201, which is the second MDM2 inhibitor from Novartis that progressed into clinical trials [42,59]. A co-crystal structure of NVP-HDM201 bound to MDM2 was obtained and showed the binding mode of this class of molecules (Figure 4C) [59]. The six-membered lactam fused to an aryl group in NVP-CGM097 was replaced in NVP-HDM201 with a five-membered lactam fused to a five-membered heteroaryl group, essentially removing the methylene group between the carbonyl and aryl, thus flattening the core and forcing the desired conformation of the *para*-chlorophenyl substituent, and so allowing the molecule to adopt the desired “central valine” binding mode (Figure 4C). In this central valine binding mode, the carbonyl of the lactam hydrogen bonds to His96, and the N-aryl substituent adopts a perpendicular conformation and becomes involved in π-π stacking with His96 of the MDM2 protein, both interactions that are not captured by the NVP-CGM907 inhibitor, which lies on the opposite side of the binding pocket [42]. Although a full report on the discovery of NVP-HDM201 has not been published to date, Novartis summarized their MDM2 program history in a short review article and indicated that NVP-HDM201 possesses advantageous physicochemical and good drug-like properties, and in their in vivo xenograft models, NVP-HDM201 was shown to be 10 times more potent than NVP-CGM097 [42]. 

Daiichi-Sankyo also pursued the discovery of novel inhibitors of the p53–MDM2 interaction and advanced DS-3032, a spiro-oxindole MDM2 inhibitor, into clinical trials [43]. No report of their discovery effort that led to this inhibitor has been published, but it is a close analog of other reported spiro-oxindoles. Unique in DS-3032 is the 2-chloro-3-fluoro-pyridine substituent at the C3 of its pyrrolidine core. All the other reported spiro-oxindoles in clinical trials contain a 2-fluoro-3-chlorophenyl substituent at this position (Figure 2). However, since no discovery process was reported, the beneficial properties of this moiety are unknown. 

During the exploration by the University of Michigan of the spiro-oxindole small molecules as p53–MDM2 inhibitors, it was discovered that the spiro-oxindoles with mono-substituted C2 of the pyrrolidine ring (now named first-generation spiro-oxindoles) suffered from equilibration through a reversible retro-Mannich recyclization reaction, which allowed the substituents to reorient into an all-trans configuration around the pyrrolidine ring leading to the major isomer [53]. Although this isomer can be isolated in high yield and is stable as a solid, the isomerization in solution caused concern over potential liabilities in development. To overcome this isomerization, the C2 was changed from a chiral center to a symmetrical center, which results in an irreversible conversion to spiro-oxindoles with a trans-configuration between the aryl substituents on the pyrrolidine ring. This yielded the second-generation spiro-oxindole class of MDM2 inhibitors, with MI-1061 as the first lead compound [60]. Further optimization efforts by bio-isosteric replacement of the benzoic acid with a bicyclo [2.2.2]octane-1-carboxylic acid resulted in an initial compound, which showed improved plasma exposure but reduced distribution in a PK study, which was reflected in its poor efficacy in a mouse xenograft model. The exploration of strategies to decrease plasma-protein binding (PPB) and improve distribution resulted in the discovery of an N-ethylated pyrrolidine spiro-oxindole MDM2 inhibitor, APG-115 (Figure 5). This compound has an MDM2 binding affinity of IC_50_ = 4.8 nM, cell activity of IC_50_ < 100 nM in wt-p53 cancer cell lines, and it retains good plasma exposure, improved distribution, and in vivo complete tumor regression in a mouse xenograft model (Table 1) [44]. Consequently, APG-115 was selected for progression to clinical trials. Interestingly, DS-3032 and APG-115 are both second-generation spiro-oxindole MDM2 inhibitors, which have a symmetrical C2 on the pyrrolidine core. No co-crystal structure of these second-generation spiro-oxindoles bound to MDM2 has been reported, but, presumably, they have a binding mode similar to that of MI-773.

Boehringer Ingelheim (BI) has also advanced an MDM2 inhibitor, BI-907828, with undisclosed structure, into clinical trials [61,62,63]. In the literature, BI has reported a novel class of spiro-oxindoles as p53–MDM2 inhibitors, and their candidate may be an optimized analog belonging to this new class of spiro-oxindoles. In this review, we will discuss insights into the reported SAR of BI-0252, which is the earliest member of this new class of reported spiro-oxindole MDM2 inhibitors [45]. Inspired by the spiro-oxindole inhibitors discovered in the Wang lab at the University of Michigan and natural product architectures, scientists at BI focused on overcoming the chemical instability of the core pyrrolidine ring of the first-generation spiro-oxindoles. Epimerization at C2 and C3 is known to occur through a reversible ring-opening, retro-Mannich reaction, and it was hypothesized that shifting the nitrogen of the pyrrolidine ring one atom closer to the oxindole would prevent this retro-Mannich reaction. Consequently, spiro-oxindoles with this modification, which effectively abolished this phenomenon, were produced [45]. Further rational design and SAR optimization to correctly substitute the pyrrolidine ring and capture the known interactions with the MDM2 pocket resulted in the discovery of BI-0252 as the lead compound. This compound has an MDM2 binding affinity of IC_50_ = 4 nM and cell activity of IC_50_ = 471 nM in SJSA-1 cancer cells (Table 1). A co-crystal structure of BI-0252 bound to MDM2 was reported and supports their design rationale (Figure 6A). Interestingly, the interactions with His-96 and Lys94 could only be captured by adding a ring, thus forming a second fused pyrrolidine ring to lock the orientation of the NH and benzoic acid substituents toward His96 and Lys94 residues in MDM2. The NH of the pyrrolidine ring in this case functions as a H-bond donor to His96; in contrast to other inhibitors, this His96 functions as a H-bond donor to a carbonyl or interacts electrostatically with a carboxylic acid. A highlight of this compound is that a high single oral dose achieved tumor regression, and the same was claimed for the clinical candidate BI-907828 [45].

The Merck Company has also worked on the discovery of novel small-molecule inhibitors of the p53–MDM2 interaction and advanced MK-8242 [47,48], a substituted piperidine class of MDM2 inhibitors [46,64,65,66], into clinical trials. Structurally, this molecule appears to be quite different from other MDM2 inhibitors, which have a more condensed core. There is no exact report of the discovery of MK-8242, but a report of a SAR study of a very close analog is available and includes its co-crystal structure bound to MDM2 [46]. This class of compounds was discovered in an in-house high-throughput screen (HTS), which identified a hit that already possessed substituents similar to those in MK-8242 but only had an MDM2 binding IC_50_ of 2.3 µM [64]. In a subsequent study, it was reported that significant improvement was achieved by adding an *n*-propyl substituent on position 2 of the piperidine core [66]. Further optimization reports indicate that replacing the trifluoromethylphenyl group, present in the initial hit, with a trifluoromethyl-thiophene moiety improved its potency, and further modification of the alkoxy substituent on the aryl ring with a long-chain carboxylic acid further improved its potency, permeability, and PK properties [46]. The co-crystal structure of this nearly optimized analog of MK-8242 showed that, like NVP-CGM097, this compound lies mainly on the wall opposite the Val93 wall on which most MDM2 inhibitors sit (Figure 6B) [46]. The piperidine core sits close to Gly58. The trifluoromethyl-thiophene effectively occupies the Trp23 pocket; the trifluoromethyl of the pyridyl group is placed deep in the Phe19 pocket; and the alkoxy phenyl group fits in the Leu26 pocket, where it π-π-stacks with His96, and its alkyl acid interacts with Gln24. Presumably, these different interactions, absent in other MDM2 inhibitors, compensate for their different binding modes, which deviate from those of other MDM2 inhibitors. The MK8242 analog has an MDM2 binding of IC_50_ = 7 nM, cell activity of IC_50_ ≤ 180 nM in wt-p53 cell lines, and it demonstrates robust in vivo antitumor activity [46], comparable to other MDM2 inhibitors (Table 1). 

Due to the high molecular weight, high clog P, and high recommended phase II dose of MK-8242, Merck has pursued lower molecular weight, novel MDM2 inhibitors. Using HTS, a purine-derived inhibitor hit was identified and optimized using the available structural information and NMR-based models [49]. Their optimization strategy focused on improving activity without significantly increasing the molecular weight by concentrating the optimization on modifying the volume, shape, and polarization of substituents that occupy each of the three well-defined binding pockets on MDM2. Their efforts yielded MK-4688, which has a MW of 550 and a predicted human dose of 75 mg QD/38 mg BID [49]. A co-crystal structure of MK-4688 bound to MDM2 was obtained, showing it efficiently captures the key Phe19, Trp23, Leu26, p53 interactions with the MDM2 protein (Figure 6C). Interestingly, this compound has a methylcyclohexyl group occupying the Trp23 pocket and is therefore different from other MDM2 inhibitors that generally use an aryl group to mimic the tryptophan residue of p53. They rationalized the use of this methylcyclohexyl group with larger hydrophobic volume to enhance contact in the pocket. Bicyclic morpholine was selected to effectively occupy the Phe19 pocket while maintaining low MW, reduced lipophilicity, and good potency to overcome its metabolic liabilities. Similarly, the 5-chloropyridine group occupying the Leu26 pocket was used to decrease lipophilicity and off-target liabilities. Finally, this molecule has only one polar interaction, an H-bond from the 1,2,4-oxadiazole substituent to the His96 of MDM2. The 1,2,4-oxadiazole was used as bio-isosteric replacement for a carboxylic acid, essentially to maintain the polar interaction with His96, and required tuning of its pKa to obtain a cell-permeable compound. Even though this molecule has only one polar interaction, the increased volume of the substituents that enhance the interactions in the key binding pockets is sufficient to achieve potent binding to MDM2 of IC_50_ = 0.65 nM, cell activity of IC_50_ = 122 nM in wt-p53 HCT116 cancer cells, and robust antitumor activity in vivo in an SJSA-1 xenograft model (Table 1). Although this compound is not currently in clinical trials, the authors indicated that it was advanced into preclinical safety studies, where it demonstrated a profile that supported its progression to the clinic.

The plethora of novel small-molecule MDM2 inhibitors discovered and advanced into clinical trials indicates the perceived importance of the p53–MDM2 axis in cancer treatment strategies. However, since p53 also retains its wild-type status in normal cells, some important limitations of MDM2 inhibitors have been observed in the clinic. For example, MDM2 inhibitors have shown hematological toxicities, mainly thrombocytopenia [67,68,69,70,71,72,73], due to activation of p53. Several strategies to circumvent this toxicity have been explored, including use of more potent compounds, which can be administered at lower doses or higher doses with less frequent administration to allow recovery [45,59,62]. Another limitation arising from activation of p53 by an MDM2 inhibitor is the upregulation of MDM2, itself a target gene of p53, as it is part of a natural autoregulation mechanism to prevent aberrant p53 activation in normal cells [27]. Hence, even though MDM2 inhibitors can strongly activate p53, the upregulation of MDM2 itself may limit their activity. 

To overcome this feedback limitation, MDM2 degraders have been designed based upon the PROteolysis Targeting Chimera (PROTAC) technology [74,75,76,77,78]. PROTAC degraders are heterobifunctional small molecules, which are composed of three structural components: a ligand for a protein of interest (POI), another ligand for an E3 ubiquitin ligase, and a linker to tether the two ligands together. A PROTAC molecule binds to the POI on one end and to an E3 ubiquitin ligase complex on the other end to bring them to close proximity to facilitate ubiquitination of the POI, leading to proteasomal degradation of the POI [79,80]. A PROTAC degrader functions as a “catalyst” in inducing the degradation of the POI, thus being capable of achieving very high degradation potency [81]. Since a PROTAC MDM2 degrader efficiently reduces the levels of MDM2, they are much more potent and effective in inducing the activation of p53. Considering these potential advantages, the Wang lab at the University of Michigan explored the PROTAC strategy for the design of MDM2 degraders. Using their own MDM2 inhibitors, they reported the first-in-class PROTAC MDM2 degraders exemplified by MD-222 and MD-224, which were designed using a potent MDM2 inhibitor MI-1061 linked to the cereblon ligand lenalidomide (Figure 7 and Figure 8) [82]. As hypothesized, Western blotting revealed complete degradation of the MDM2 protein by its MDM2 degraders at very low concentrations, leading to robust increase in p53 protein and increased expression of p53-targted genes and gene products and apoptosis induction in p53 wild-type tumor cells. PROTAC MDM2 degraders are 100 times more potent than their corresponding MI-1061 inhibitor at inhibition of cell growth. MD-224 achieves much stronger antitumor activity in vivo at much lower doses and lower dosing frequency than its corresponding inhibitor. Further optimization of this class of PROTAC–MDM2 degraders yielded AA-265, which is currently in advanced preclinical evaluation for progression into clinical trials.

Notably, during the exploration of PROTAC–MDM2 degraders, it was discovered that removing the benzamide substituent from the MDM2 inhibitor portion resulted in MG277, which showed increased potency even in mutant or deleted p53 cancer cells. Further research revealed that this structural modification converted the chimera into a molecular glue, which degraded the G1 to S phase transition 1 (GSPT-1) protein, a translation termination factor, also known as eRF3a, which mediates stop codon recognition and nascent protein release from the ribosome through the interaction with the release factor, eRF1 (Figure 8) [83]. Thus, the activity in cell lines with compromised p53 was due to GSPT-1 degradation and not to MDM2 degradation. Interestingly, compounds with benzamide or terminal amides, such as MD-222 and MD-224, were all selective MDM2 degraders and did not have off-target GSPT-1 activity [83].

Shortly after the first PROTAC–MDM2 degraders were disclosed by the Wang group at the University of Michigan, the Tang group at the University of Wisconsin-Madison reported the discovery of a PROTAC–MDM2 degrader derived from the Nutlin class of MDM2 inhibitors and the cereblon ligand lenalidomide [84]. The characteristic feature of these PROTAC degraders is that the linker-lenalidomide moiety is tethered from the solvent projecting piperazine substituent of the Nutlin class of MDM2 inhibitors. This class of compounds is exemplified by WB156 (Figure 7), which boasts the shortest linker ever reported and achieves > 1000-fold improvement in cell potency relative to its corresponding inhibitor [84]. 

Due to the activity of WB156 on a limited number of leukemia cells, Tang et al. explored the creation of a new class of PROTAC–MDM2 degraders derived from another class of known MDM2 inhibitors, which could be diversified by a four-component Ugi reaction. This produced a new class of heterobifunctional compounds, including WB214 (Figure 7), which displayed the best potency and activity in a broader group of cancer cells [85]. Interestingly, the analysis of its mechanism of action (MOA) revealed that this new class of compounds also functioned as a GSPT-1 degrader, which was independent of MDM2 and p53 degradation. They found that p53 was degraded as a bystander, which is associated with MDM2, and thus suggested that this class of molecules may bind to other parts of MDM2 and not the same pocket occupied by p53 [85].

Two other PROTAC–MDM2 degrader molecules have been reported, but their structures have not been disclosed. Marcellino et al. reported the development of the PROTAC–MDM2 degrader XY-27 using an undisclosed MDM2 inhibitor tethered to a ligand of the VHL E3 ubiquitin ligase [86]. VHL was selected due to its higher expression in acute myelogenous leukemia (AML) compared to other cancers, and its use can potentially improve the potency and specificity in AML. The VHL-based PROTAC–MDM2 degrader demonstrated superior potency over the AMG232 MDM2 inhibitor in MOLM13 and MV4;11 cell lines and showed potent activity when combined with azacytidine or cytarabine. Kymera Therapeutics also reported the development of a PROTAC–MDM2 degrader KT-253 using an undisclosed MDM2 inhibitor tethered to a ligand of an undisclosed E3 ubiquitin ligase [87]. They report picomolar DC_50_ results (Table 2) and an cell killing activity, which is >200-fold more potent than small-molecule inhibitors of MDM2 in clinical trials and especially compared to DS-3032. Impressively, a single dose of KT-253 at 1, 3, or 10 mg/kg achieved complete tumor regression in an RS4;11 xenograft model in mice and was longer lasting with the 10 mg/kg dose. Kymera Therapeutics indicated that they intend to move this compound into clinical trials, with anticipated IND filing in 2022 [87].

These MDM2–PROTAC reports indicate that application of the PROTAC strategy for targeted degradation of MDM2 is indeed achievable; however, the molecular glue activity seen in some putative MDM2 degraders indicates that if other MDM2 inhibitors are used to produce PROTAC–MDM2 degraders, a careful profiling of the MOA is necessary to define the mechanism of action. Of significance, bona fide PROTAC–MDM2 degraders have been shown to overcome the MDM2–p53 feedback loop, leading to more robust antitumor activity in vivo than MDM2 inhibitors.

### Disrupting MDMX–p53 Protein–Protein Interactions

In some cancers, such as retinoblastomas [88], breast carcinomas [89], and melanomas [90], the inhibition of MDM2 may not be sufficient to stop tumor progression because the cancers highly express MDMX, which is a structural homolog of MDM2. MDMX does not have E3 ligase activity but still participates in the regulation of p53 by binding to the N-terminus of p53 and thus neutralizing its transactivation activity [91]. In addition, MDMX and MDM2 share a comparable sequence in their RING domain, and through their domains, MDMX binds to MDM2 [92,93]. This stabilizes MDM2, resulting in the potentiation of MDM2′s capability for ubiquitination of p53 [94,95]. Furthermore, the same p53 residues that bind to MDM2 also bind to a similar cleft in MDMX, and this indicates that the inhibition of the MDMX–p53 interaction is plausible and may be possible and beneficial for cancer treatment [34,96]. Below, we summarize the progress in the efforts to inhibit the MDMX–p53 interaction, focusing only on the compounds that have been shown to have reasonable binding to MDMX or have co-crystal structures that can provide insights for targeting MDMX (Figure 9).

Due to the high similarity of the MDMX and MDM2 binding pockets, the same p53 residues Phe19, Trp23, and Leu26 also bind to MDMX, indicating that selective MDMX inhibition will be challenging [34,96]. Hoffmann-La Roche, after abandoning RO-5963, a small-molecule dual inhibitor of MDMX/MDM2 (discussed below), due to its poor pharmacological characteristics [8], approached dual inhibition by optimization of reported short peptides, which individually potently bind to MDMX/MDM2. It had been shown that appropriate stapling of short peptides resulted in stabilization of the α-helical conformation, improved permeability, and other physicochemical properties, and as a result, it became the optimization strategy employed by Hoffmann-La Roche [97]. Their optimization efforts focused on improving biological and physicochemical properties, and this effort provided ATSP-7041 [97]. Subsequently, Aileron Therapeutics reported ALRN-6924 [98], which is currently in clinical trials (Figure 9). ATSP-7041 is a potent dual inhibitor, which is cell permeable and boasts significantly slow off-rate kinetics compared to small-molecule inhibitors and causes an extended inhibitory effect even in cells. Additionally, its optimized α-helical stability resulted in improvement in cell permeability, in vivo stability, favorable PK properties, and, consequently, potent cellular and in vivo antitumor activity in cancers that overexpress MDM2 or MDMX [97]. The advantages of ATSP-7041 and ALRN-6924 dual inhibition of MDMX and MDM2 were shown by head-to-head comparison with selective MDM2 inhibitors, RG7112 and RG7388, in cancers that overexpress MDMX, where these dual-peptide-based inhibitors showed superior activity [97,98]. These results support the idea that dual inhibition of MDMX and MDM2 may be superior to selective inhibition of only one of these homologs. No co-crystal structure of ALRN-6924 bound to MDMX or MDM2 has been reported, but a similar stapled peptide ATSP-7041 was successfully co-crystallized with MDMX and showed the binding mode of these stapled peptides (Figure 10A) [97]. This structural information indicates that the natural p53 residues, Phe19, Trp23, and Cba26 (a cyclobutyl Leu26 isostere), present in ATSP-7041, occupy the same p53 binding pockets in MDMX. In addition, its Tyr22 residue captures additional hydrophobic interactions and a water-mediated hydrogen bonding interaction with Lys93 and His68. The aliphatic stapled chain also captures additional hydrophobic interactions and potentially a cation–π interaction between its alkene and His51 on the MDMX surface wall. These interactions are the structural basis for the strong binding of these stable stapled peptides to both MDMX and MDM2.

It is conceivable that if the same residues of p53 are critical for binding to both MDMX and MDM2, then the small molecules that mimic them and have been used as MDM2 inhibitors should also achieve potent MDMX inhibition. Interestingly, this is not the case, and it has been seen that many potent MDM2 inhibitors do not have potent binding to MDMX [99,100]. The enhanced activity observed from the peptide-based dual inhibitors in cancers, which carry high levels of MDMX, supports the development of small-molecule selective or dual MDMX/2 inhibitors, and efforts toward this end will be summarized here—specifically for compounds with confirmed binding to MDMX with co-crystal structures. 

One of the first small molecules that showed nM binding to MDMX and was also equipotent to MDM2 is RO-2443 (Figure 9) [8]. This dual inhibitor was discovered by Hoffmann-La Roche through high-throughput screening for MDMX binding and initially had a predicted binding mode in which its indole moiety fit into and mimicked the Trp23 of p53. However, NMR and size exclusion chromatography studies indicated MDMX dimer formation, and the co-crystallization efforts confirmed an MDMX dimer that sandwiches two molecules of RO-2443 (Figure 10B) [8]. In the dimer crystal structure, the indolyl-hydantoin moieties of each RO-2443 π-π stack on top of each other, and each one occupies an extended Phe19 pocket of one of the MDMX protein monomers. Interestingly, the difluorobenzyl substituent occupies the Trp23 pocket in the opposite MDMX protein monomer, indicating that, essentially, two molecules are required to complement their binding in the MDMX dimers, and this is necessary if potent MDMX and MDM2 binding is to be achieved. Due to the poor solubility of RO-2443, it was not suitable for intracellular evaluation; therefore, its more soluble analog, RO-5963, was synthesized with a diol containing carboxamide substitution at the methylene position of the benzyl group (Figure 9) [8]. This substitution was reported to potentially capture additional interactions in the unoccupied Leu26 pocket and to improve its solubility for evaluation in cell-based assays. In fact, this compound demonstrated superior cell activity over Nutlin-3a in cancer cells that overexpress MDMX. This indicates that dual MDMX/MDM2 inhibitors may be more suitable in cancers that have high levels of MDMX. Its poor pharmacological characteristics prevented the further development of RO-5963 [8].

Although binding to MDMX only weakly, the MDM2 inhibitor WK298 (Figure 9) was the first non-peptide small molecule, which was successfully co-crystalized with MDMX. This showed unambiguously that small molecules that mimic the three key p53 binding residues, Phe19, Trp23, and Leu26, can indeed bind to both MDM2 and MDMX (Figure 10C) [101]. The phenyl, chloroindole, and 4-chlorobenzyl substituents of WK298 effectively occupy the Phe19, Trp23, and Leu26 p53 binding pockets of MDMX, respectively, similar to their binding modes in MDM2. However, in the case of MDMX, these interactions are not sufficient for potent binding and highlight the challenge for the development of potent MDMX inhibitors. This lack of improved binding was attributed to the residues, which are different in MDM2 and MDMX, especially in the Leu26 pocket, which contains most of the differences [101]. In this pocket, the MDM2 residues Leu54, Ile103, Ile99, and His96 are the Met53, Leu102, Leu98, and Pro95 residues in MDMX, respectively. This change makes the MDMX pocket more open than the pocket in MDM2 and presumably decreases the tight binding. Other MDM2 inhibitors have also been co-crystallized with MDMX, but they did not reveal any additional interaction, which is unique to MDMX, and thus failed to achieve improved MDMX inhibition [99].

A more highly substituted analog of WK298, called Novartis-14 (Figure 9), was found to potently bind MDMX with a TR-FRET IC_50_ of 17 nM (Table 3), and the co-crystal structure of this compound reveals additional interactions that underlie its potent MDMX binding (Figure 10D) [99]. As in WK298, the substituted imidazole core in Novartis-14 effectively occupies the Phe19, Trp23, Leu26 p53 binding pockets in MDMX, but additional interactions add to the tighter binding to MDMX. The 2,4-dichlorobenzyl substituent in Novartis-14 can occupy the more open Leu26 p53 binding pocket in MDMX, and the large amide substituent that hovers above the core projects its hydrophobic oxazinanone into the expanded Phe19 p53 binding pocket in MDMX. These two both contribute to the improved binding to MDMX. More importantly, the carboxylic acid of its benzoic acid substituent is positioned effectively close and captures a strong electrostatic interaction with the His54 residue of MDMX, another of the residues, which distinguishes it from the MDM2 (Phe55) binding pocket. This electrostatic interaction is important for MDMX binding, as its analogs, which lack or have a blocked carboxylic acid, significantly suffer from weaker MDMX binding, as seen in the related patent [102].

These reports indicate the potential for obtaining selective MDMX or dual MDMX/MDM2 inhibitors. From the Novartis-14 data, it was revealed that capturing an interaction with the His54 residue of MDMX improved its binding. The crystal structures also revealed that both the Phe19 and especially the Leu26 binding pocket in MDMX are more open, indicating that targeting these pockets may be a potential approach to tuning in MDMX binding and design of selective inhibitors. Additionally, no PROTAC–MDMX degraders have been reported, and the discovery of these MDMX binders may provide an opportunity for exploration of this strategy. To the best of our knowledge, no non-peptide small-molecule dual MDMX/MDM2 inhibitor has advanced to clinical trials, but certainly, these data will give useful insights into improvements of MDMX binding, which can be used in programs with this target.

## 3. Mutant p53 Binders as Mutant p53 Activators

Reactivating wt-p53 by targeting its negative regulators, MDM2 and MDMX, has seen significant progress. However, approximately 50% of human cancers harbor mutated p53, and both MDM2/MDMX inhibitors and MDM2 degraders are ineffective in inducing activation of mutated p53 [103,104,105,106,107]. Mutations in p53 typically result in loss of the wt-p53 function and, in some cases, even in gain of the oncogenic function [108,109]. The loss of function results from the mutant p53 being unable to bind to DNA either by loss of residue contact to DNA or loss of the ability to fold into its active conformation required for binding to DNA. Interestingly, some cancers with mutant p53, particularly early-stage cancers, are heterozygous, and the mutant p53 dimers form an inactive heterotetramer with wt-p53 dimers, acting as dominant negatives and neutralizing the wt-p53 activity [20,110,111]. On the other hand, mutant p53 can have a gain of function by interacting with other partners, such as the transcription factors p63 and p73, blocking their anticancer activity [112,113]. Furthermore, mutant p53 can also bind to other transcription factors or cofactors to enhance other genes, resulting in tumor cell survival [113] (Figure 11).

Direct targeting of the p53 protein is therefore a very appealing anticancer therapeutic strategy. Unfortunately, the p53 protein is frequently considered to be “undruggable” because neither wt-p53 nor mutant p53 has binding pockets or allosteric sites, which can be readily targeted by small molecules [23]. This is further complicated by the existence of over 2000 possible mutant p53 variants. However, recent progress has been made in restoration of wt-like activity in mutant p53 [23,107,114,115]. Here, we summarize some of the advances in this research. 

Of the >50% of human cancers that have *TP53* gene mutations, 75% are missense mutations and mainly occur in the DBD of p53 [116,117,118]. Most of these mutations occur in “hot-spot” residues, and the change in these residues results in either loss of contact with DNA or change in the structural stability of the p53 protein (Figure 11) [23,119,120]. Consequently, the mutants can be divided into contact mutants or structural mutants [2,23,113,121,122,123]. Contact mutants involve residues that interact directly with DNA, and for effective targeting, the introduction of interactions to compensate for missing DNA contacts is necessary [124]. Due to the lack of well-defined pockets in these contact p53 mutants, targeting them with small molecules can be very challenging [121]. On the other hand, structural mutants have residue mutations, which influence the thermal stability and structure of p53, making the mutants unstable at physiological temperatures and preventing them from achieving correct folding and having wt-like functions [119]. Therefore, small molecules, which can bind, stabilize, and restore the level of correctly folded p53, may represent a more achievable strategy in targeting structural p53 mutants. Indeed, several small molecules that stabilize an active p53 conformation and restore wt-like transcriptional function of mutant p53 have been reported [23,107,114,115].

Several reviews on small molecules targeting mutant p53 have been published, but the MOA for some of these small molecules is not fully understood. They can sometimes rescue both contact and structural mutants of p53 but may also have p53-independent activity [23,119]. Here, we focus on those molecules, which have crystal structural information supporting their direct binding to mutant p53. Among these are some small molecules that target the hydrophobic pocket created at the mutation site of the structural mutants of tyrosine 220 (Y220). Small molecules have been designed to target this site and were successful in reactivation of the wt-like p53 function of these p53 mutants. Additionally, the co-crystal structures of these small molecules bound to the Y220C mutant have shown unambiguously that this mutant pocket can be targeted with small molecules. Targeting mutant p53 to rescue its wt-like function has yielded only two molecules, which have entered clinical trials, PRIMA-1^MET^(APR-246) [125] and COTI-2 [126,127].

APR-246 is a more potent methylated derivative of PRIMA-1 and was discovered by screening for compounds that show mutant-p53-dependent cell growth inhibitory activity and also reactivate mutant p53 (Figure 12) [125,128]. It was found that these compounds restore an active p53 conformation, which can bind to DNA and consequently rescue wt-like transcription activity, which results in induction of apoptosis in human cancer cells. It was found that APR-246 and its parent PRIMA-1 are prodrugs, which are converted to methylene quinuclidinone (MQ)—a very reactive Michael acceptor, which reacts with cysteine residues in “the core domain” of mutant p53 (Figure 13) [129,130]. The exact mechanism of the reactivation of p53 by APR-246/MQ is not fully understood; however, a study of Cys to Ala mutations in the p53 core domain using mass spectrometry identified Cys277 as the most nucleophilic and most solvent-accessible of the 10 cysteines in the core domain. Further studies showed that Cys277 and Cys124 both have an important role in APR-246/MQ-mediated mutant p53 reactivation [129]. Similarly, an independent study using various high-resolution crystal structures of mutant and wt-p53 core domains bound to MQ also identified the same Cys124 and Cys277, and, in addition, Cys229, as contributors to MQ-mediated stabilization of mutant and wt-p53 [131]. These studies concluded that these alkylating agents react with the most exposed cysteine residues in the p53 protein. MQ was shown to be able to rescue wt-p53-like function in both structural and contact mutants. Interestingly, it was also found that APR246/MQ also induces high levels of reactive oxygen species resulting from its depletion of glutathione (GSH), contributing to their anticancer activity [132,133].

The clinical compound COTI-2 is a third-generation thiosemicarbazone class molecule, which was identified using CHEMSAS^®^, a proprietary computational platform that incorporates various drug discovery principles and technologies to optimize novel compounds, which could target various human malignancies [134]. COTI-2 was found to be active against a wide variety of human cancer cell lines and xenografts, which are traditionally difficult to treat; however, its mechanism of action was not known at the time. The initial studies suggested that the MOA of COTI-2 involved negative modulation of the PI3K/AKT/mTOR pathway. COTI-2 was evaluated in a variety of cancer cell lines encompassing various types of human malignancies, including cancers with *TP53*, *KRAS*, *PIK3CA*, *APC*, and *PTEN* mutations. While COTI-2 demonstrated potent activity in all of these cancer cell lines, it was especially potent against mutant p53 cancer cell lines [134]. Subsequently, it was reported that COTI-2 restored wt-like p53 function through a mechanism involving zinc chelation, in addition to modulation of the PI3K/AKT/mTOR pathway [135]. While the exact mechanism of COTI-2 is unclear, its potent activity and low toxicity in a variety of difficult-to-treat cancer models has resulted in its progression into human clinical trials [134].

Structural mutants of p53 result in reduced thermal stability of the DBD, causing it to unfold and aggregate at physiological temperatures, thus becoming inactive [120,136,137]. It is estimated that one-third of the p53 mutants have lower melting temperatures, but studies have shown that at temperatures below physiological temperatures, their wt-like conformations are restored. This indicates that if a small molecule can bind to mutant p53 and increase the folded thermal stability of mutant p53, it may also restore wt-like conformation and function of mutant p53 [137]. This approach has been pursued, and one of the most attractive of these structural mutants is Y220C, which is found annually in about 100,000 new cancer cases [138]. This mutation results in a highly destabilized mutant p53 protein and is also known to create a pocket that is distant from the surface involved with DNA binding and is thus an attractive site for targeting by small molecules [138].

A group at the Medical Research Council, Cambridge, UK, reported a class of carbazoles exemplified by PK083 (Figure 12) as activators of mutant p53 Y220C [139]. This was accomplished by in silico analysis of the T-p53C-Y220C crystal structure by virtual screening, followed by NMR screening of the hits, examining changes in the chemical shift of ^15^N-labeled-T-p53C-Y220C or T-p53C. This molecule stabilized the wt-like folded conformation of this mutant, and it was shown unambiguously by a high-resolution co-crystal structure that PK-083 bound in the Y220C pocket of p53. Further optimization efforts by the same group led to the carbazole analog PK-9328 (Figure 12 and Figure 14A), whose 4-methylthiazole ring was designed to occupy the pocket formed by Pro152 and Pro153, enabling PK-9328 to improve the K_d_ by >70-fold from 125 µM in PK-083 to 1.7 µM in PK-9328 [137]. Similarly, the binding of PK-9328 to the Y220C crevice and stabilization of the mutant p53 were demonstrated using biophysical and cell-based techniques and by its co-crystal structure bound to the Y220C-p53 mutant protein, confirming the additional interactions in the Pro152, Pro153 pocket (Figure 14A) [137].

Using the same strategy, the Cambridge (UK) group reported the discovery of three other classes of Y220C-p53 mutant activators, which function by binding to the Y220C pocket. The 2-(aminomethyl)-4-ethynyl-6-iodophenols exemplified by PK-5196 [140], the substituted 4-phenyl-3-(1H-pyrrol-1-yl)-1H-pyrazole class exemplified by the soluble analog PK-7242 [141], and the aminobenzothiazole derivatives exemplified by MB-710 [142] were all shown by biophysical and cell-based methods to stabilize the Y220C-p53 mutant, and the co-crystal structures showed that they all bind in the Y220C pocket (Figure 14B–D). In comparison, their crystal structures showed that the cores of these three molecules are mainly sandwiched between Val147 and Pro151 on one side, and Pro222 and Pro223 on the other side of the pocket, and similarly to PK-9328, they all have substituents that extend into the pocket surrounded by Pro152 and Pro153 (Figure 14A,B). However, these three molecules do not extend so far toward the Trp146 side of the pocket. In fact, their substituents, which face this side of the pocket, have structural water-mediated polar interactions with backbone carbonyls of Val147, Leu145, and Asp228 (Figure 14B–D) [140,141,142]. In PK-9328, one of the aryl rings displaces these structural water molecules, and the methylenemethylamine chain, which is extended from this aryl ring, forms a hydrogen bond with the backbone carbonyl of Asp228 (Figure 14A) [137]. This additional occupancy of the pocket may be the reason why PK9328 has a better binding K_d_ than the other three classes of compounds. Even considering the better binding K_d_ of PK-9328, all these small molecules still only have µM potency, which may be inadequate for drug candidates. However, the significant improvement in potency compared to the initial lead compound, PK-083, indicates that there may still be opportunity for improvement. Potentially, this can be achieved by capturing additional hydrophobic and polar interactions with the surface residues, such as those seen with the structural waters, or by the selective covalent binding to the Cys220 residue, which is unique to this mutant and forms part of the pocket.

In their continued pursuit of mutant p53 Y220C binders and activators, the same group discovered the 2-sulfonylpyrimidine PK-11000 series (Figure 12) [143]. Through their NMR, mass spectrometry, and crystallographic studies, they discovered that this compound did not bind in the Y220C crevice but instead added covalently to the two most reactive and exposed cysteine residues, Cys182 and Cys277, and this addition was selective even at high concentrations. Although PK-11007, a more reactive analog, was found to increase the thermal stability of mutant p53 and restore the wt-like activity by upregulation of p53 target genes, such as p21 and PUMA, in some cells, the major cell death activity was found to implicate two other mechanisms, namely, induction of high levels of reactive oxygen species, which resulted from glutathione depletion, and also an unfolded protein response (UPR) triggered by endoplasmic reticulum (ER) stress, both of which are similar to that observed in APR-246 [143]. In their crystallization efforts, high-resolution data are available only for the Cys182-PK-11000 adduct and confirm the structure of the S_N_Ar adduct (Figure 15). In the crystal structure of this adduct, the alkylating agent is not located in any crevice, signifying that the covalent bonding is not facilitated by other interactions with the protein. Instead, the conclusion supports the finding that alkylating agents probe and react with the most exposed and reactive cysteines, among which Cys182 and Cys277 are the most reactive in the DBD. This study indicates that alkylating agents may not have an exclusive MOA of p53 alkylation but may also be subject to consequences of alkylation of other reactive thiol-containing protein targets.

These studies have shown that the structural mutant Y220C, which creates a pocket in mutant p53, can be successfully targeted by various classes of small molecules. Importantly, the binding of small molecules in this pocket results in stabilization of the wt-like conformation and rescues the wt-like activity of this mutant. Recently, PMV Pharma reported PC14586, a small molecule whose structure is undisclosed but which binds specifically to the Y220C pocket and is the first-in-class Y220C reactivator to enter clinical trials [144]. They report tumor regression in xenograft models of Y220C p53 mutant tumors and indicate an IC_50_ of ~192–722 nM in cell lines generated from Y220C mutant lymphoma and sarcoma tumors (Table 4). In addition, they showed a synergistic antitumor effect when the Y220C activator was combined with anti-PD-1 checkpoint therapy and are currently investigating this in clinical trials [144].

## 4. Conclusions

The tremendous efforts at reactivation of wild-type and mutant p53 clearly demonstrate their unmistakable importance as novel cancer therapeutic strategies. Successful activation of wt-p53 through disruption of its principal negative regulators, MDM2 and MDMX, has yielded several molecules, which are being clinically evaluated and have shown some success. However, this accounts for only half of all cancers, and in the other half, p53 is compromised by mutations that can also occur from treatment with MDM2 inhibitors. Ideally, it would be beneficial to achieve similar success in the reactivation of mutant p53, but this has been more challenging than in the MDM2/MDMX field. The progress seen in targeting the p53 mutant Y220C with small molecules, along with their structural information and successful rescuing of wt-like functionality, provides the proof of concept and a tremendous opportunity to develop a new class of anticancer drugs for this mutant, which has a high rate of incidence in human cancers. It is exciting that the Y220C binder PC14586 has recently progressed into clinical trials.

Activation of wt-p53, through targeting the MDM2/MDMX–p53 interactions, and rescue of wt-like activity of Y220C mutant p53, through selective mutant Y220C binders, have successfully yielded molecules, which have advanced to clinical trials. Both have shown that, in fact, wild-type and mutant p53 activation leads to powerful anticancer activity, albeit with some limitations. Of the clinical trials in progress, with the exception of those of single agents, the majority are studies in combination with other therapeutic agents, which, preclinically, have shown synergy and the ability to potentially circumvent the limitations by attacking related and unrelated mechanisms, giving cancer cells less chance to develop resistance. These clinical trials are summarized in Table 5. Although none of these programs has an approved compound, the activation of p53 is still an attractive cancer therapeutic strategy.

## Figures and Tables

**Figure 1 pharmaceuticals-16-00024-f001:**
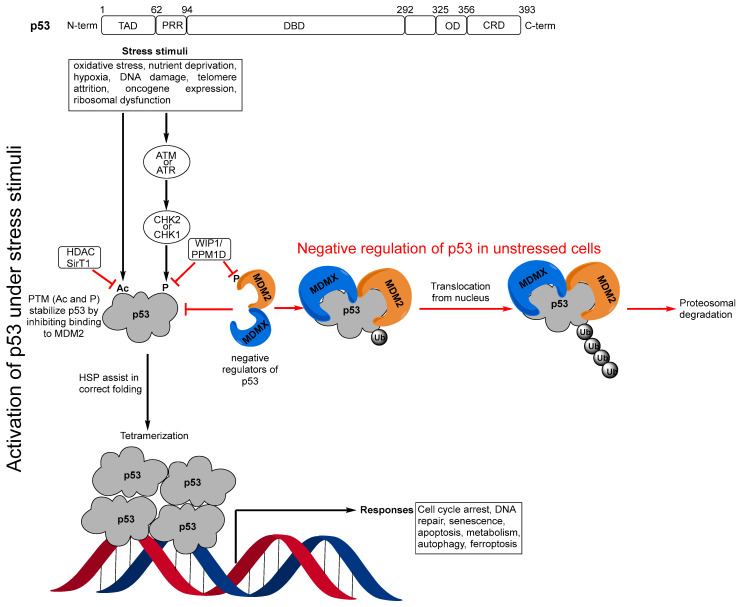
(p53 domains) Transactivation domain (TAD), proline rich region (PRR), DNA binding domain (DBD), oligomerization domain (OD), C-terminal regulatory domain (CRD); (Negative regulation of p53 in unstressed cells) In unstressed conditions, p53 levels are kept low by binding to its major negative regulators MDM2 and MDMX, which block its transactivation activity and polyubiquitinate it for proteosomal degradation; (Activation of p53 under stress stimuli) Under stress stimuli, the post-translational modifications (PTM), acetylation (Ac), or phosphorylation (P) by checkpoint kinase 1 or 2 (CHK1 or CHK2), which result from activation of the upstream kinases ataxia telangiectasia mutated (ATM), and ataxia telangiectasia and Rad3-related protein (ATR), stabilize and activate p53 by inhibiting its binding to MDM2. Heat shock proteins (HSP) then assist in the correct folding of the p53 monomer, which then forms its active tetrameric form, which binds to DNA for gene transcription. In this activation process, there are also other mechanisms that can result in repression of p53 through removal of the PTMs. The histone deacetylase (HDAC), sirtuin 1 (SirT1), deacetylates p53 and enhances its ubiquitination by MDM2. p53 also induces expression of another of its negative regulators, WIP1/PPM1D, which is a phosphatase 1 that destabilizes p53 by dephosphorylating serine 15 in p53 but also by dephosphorylating MDM2, resulting in stabilization of MDM2.

**Figure 2 pharmaceuticals-16-00024-f002:**
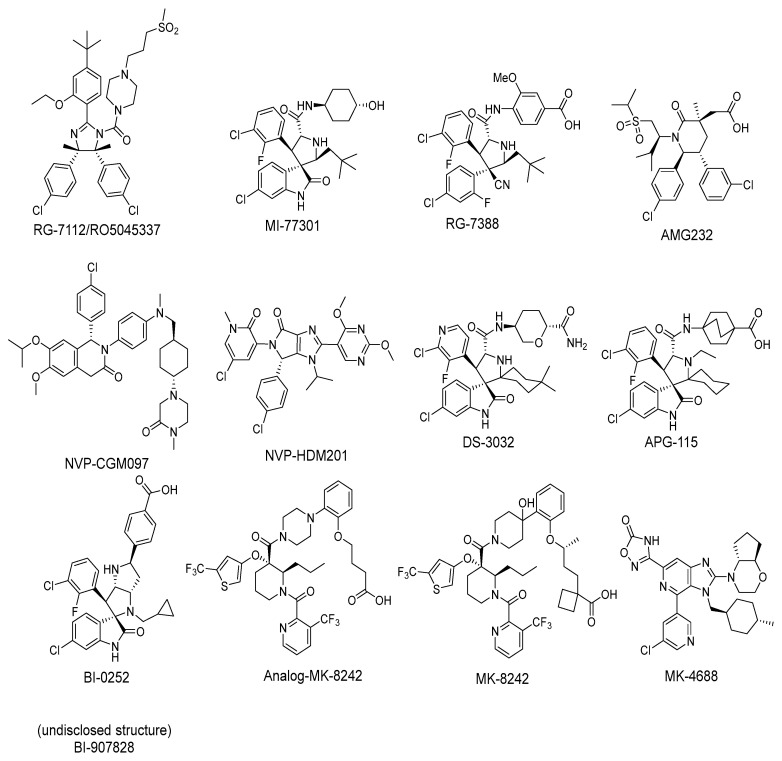
Chemical structure of MDM2 inhibitors.

**Figure 3 pharmaceuticals-16-00024-f003:**
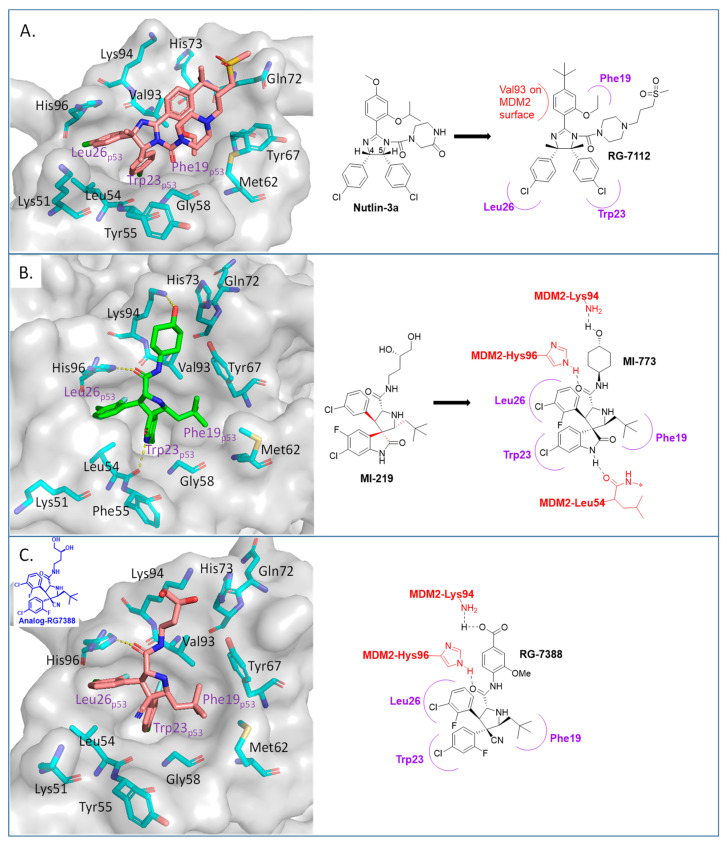
Co-crystal structures and chemical structures of inhibitors bound to MDM2 protein: (**A**) RG7112; (**B**) MI-773; and (**C**) RG7388. Labels and semicircles in magenta indicate the binding pockets in MDM2 where the p53 residues Phe19, Trp23, and Leu26 bind. Labels in red indicate MDM2 residues.

**Figure 4 pharmaceuticals-16-00024-f004:**
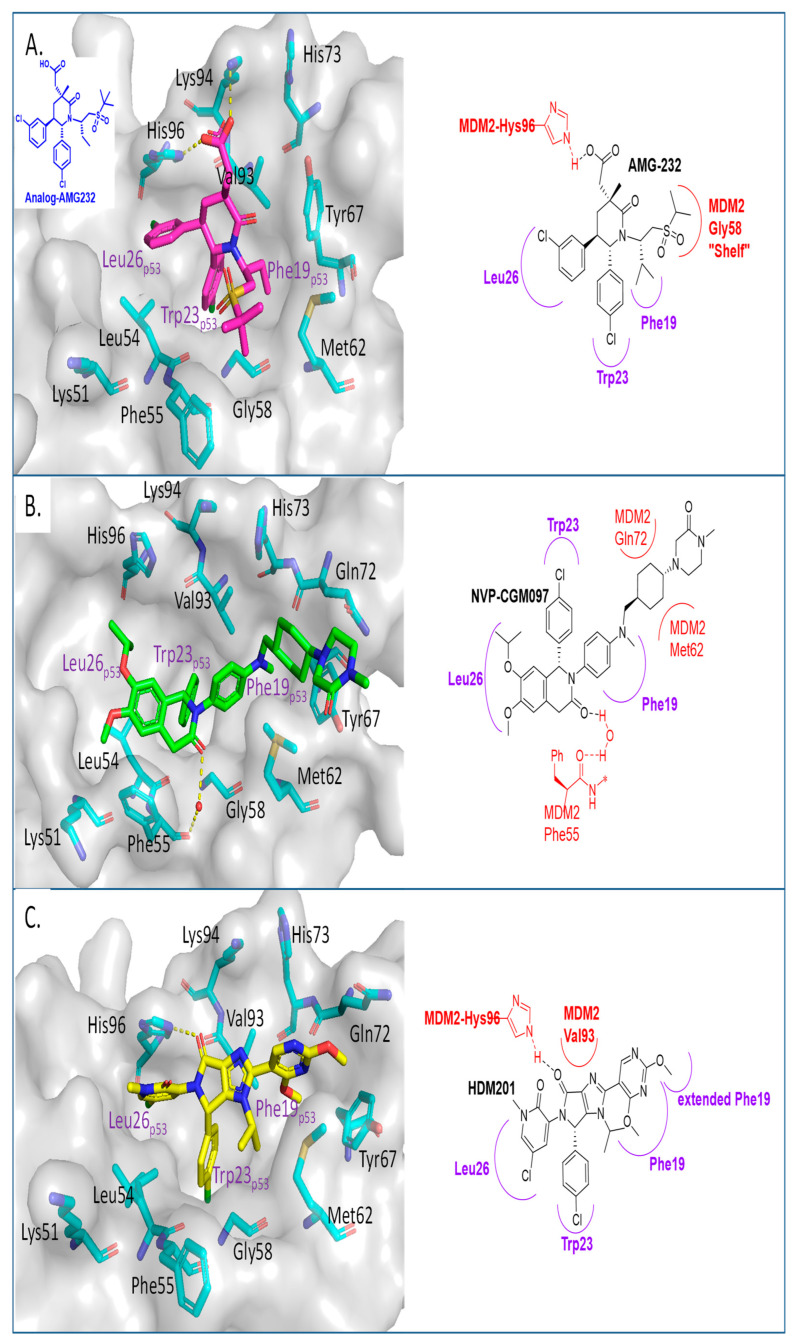
Co-crystal structures and chemical structures of inhibitors bound to the MDM2 protein: (**A**) AMG-232; (**B**) NVP-CGM097; (**C**) and NVP-HDM201. Labels and semicircles in magenta indicate the binding pockets in MDM2 where the p53 residues Phe19, Trp23, and Leu26 bind. Labels in red indicate MDM2 residues.

**Figure 5 pharmaceuticals-16-00024-f005:**
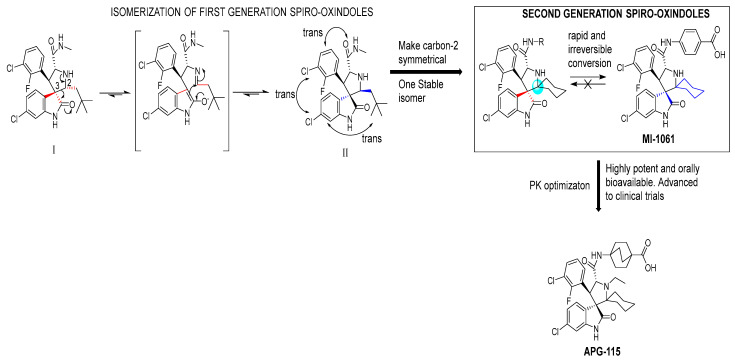
Discovery of second-generation spiro-oxindole class of MDM2 inhibitors. Red lines indicate the starting stereochemistry and the blue lines indicate the desired ending stereochemistry. The blue circle indicates the symmetrical stereocenter in the second-generation spiro-oxindoles.

**Figure 6 pharmaceuticals-16-00024-f006:**
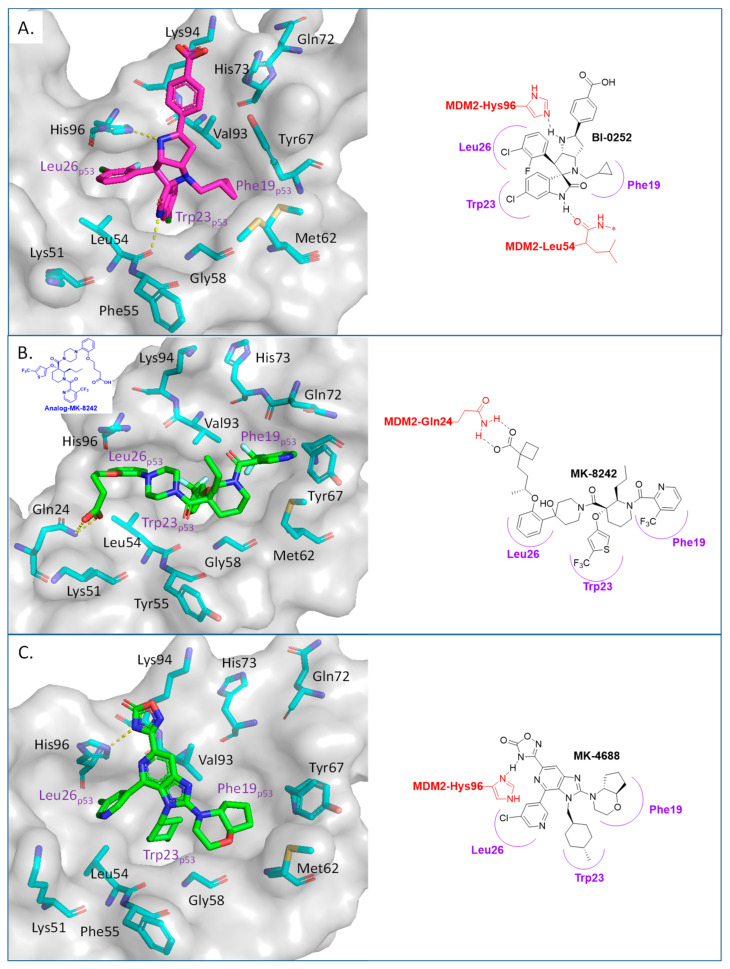
Co-crystal structures and chemical structures of inhibitors bound to MDM2 protein: (**A**) BI-0252; (**B**) MK-8242; (**C**) and MK-4688. Labels and semicircles in magenta indicate the binding pockets in MDM2 where the p53 residues Phe19, Trp23, and Leu26 bind. Labels in red indicate MDM2 residues.

**Figure 7 pharmaceuticals-16-00024-f007:**
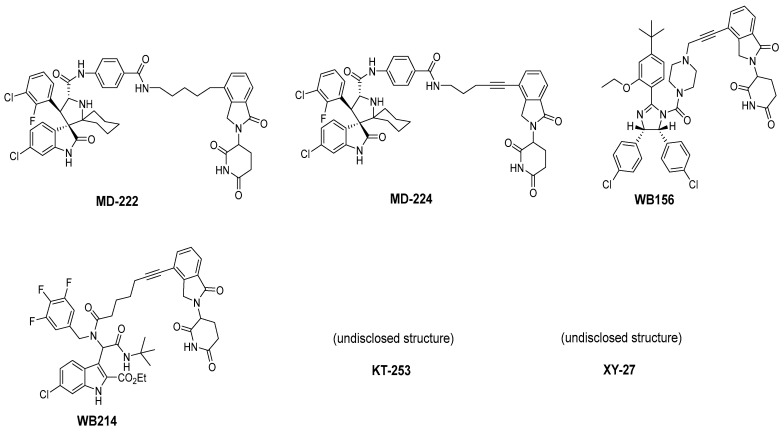
Chemical structure of PROTAC–MDM2 degraders.

**Figure 8 pharmaceuticals-16-00024-f008:**
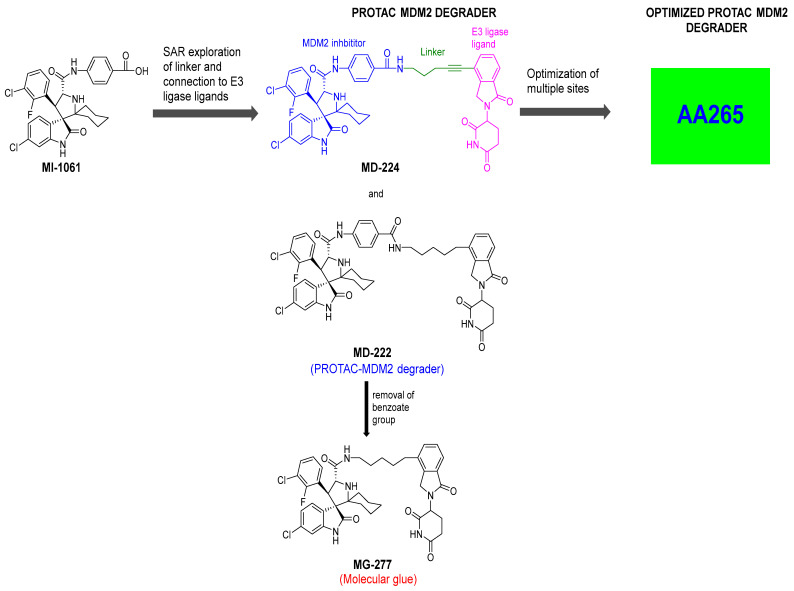
Discovery of PROTAC–MDM2 degrader and molecular glue that degrades GSPT1.

**Figure 9 pharmaceuticals-16-00024-f009:**
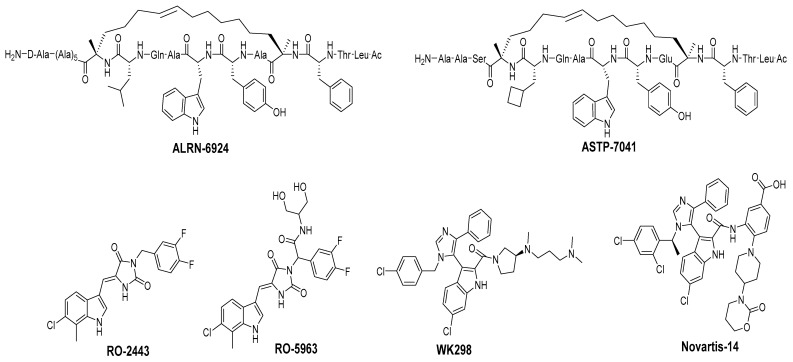
Chemical structure of MDMX inhibitors.

**Figure 10 pharmaceuticals-16-00024-f010:**
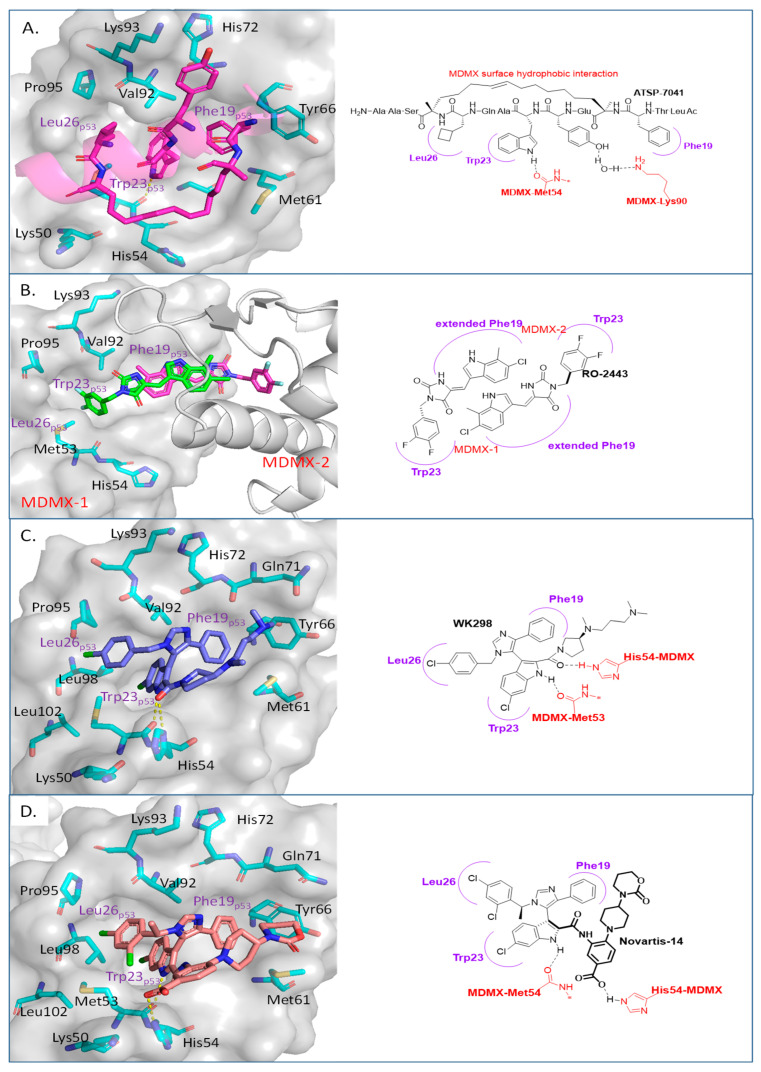
Co-crystal structures and chemical structures of inhibitors bound to MDMX protein: (**A**) ATSP-7041; (**B**) RO-2443; (**C**) WK298; (**D**) and Novartis-14. Labels and semicircles in magenta indicate the binding pockets in MDMX where the p53 residues Phe19, Trp23, and Leu26 bind. Labels in red indicate MDMX residues.

**Figure 11 pharmaceuticals-16-00024-f011:**
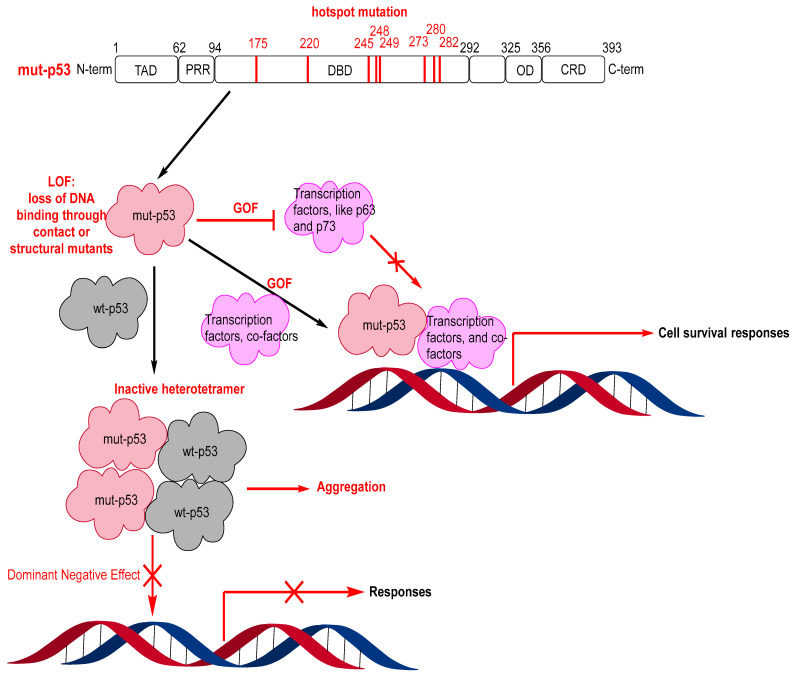
Mutant p53 hotspot mutations, loss of function (LOF), gain of function (GOF), and dominant negative effect (DNE).

**Figure 12 pharmaceuticals-16-00024-f012:**
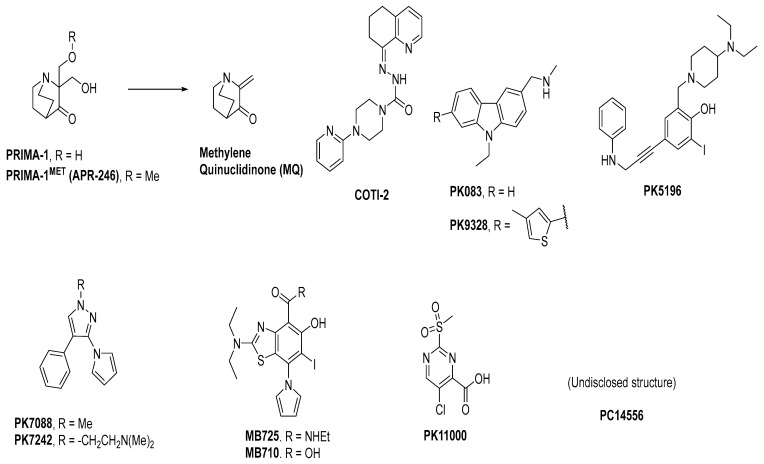
Mutant p53 activators.

**Figure 13 pharmaceuticals-16-00024-f013:**
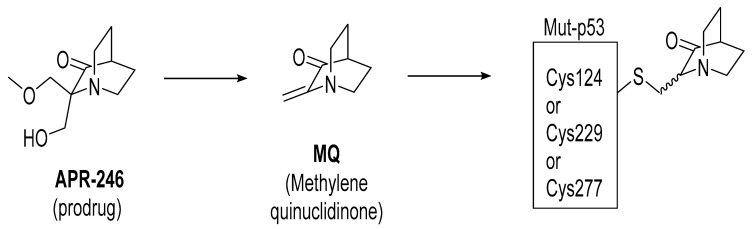
APR-246 forms MQ, which is a reactive Michael acceptor, which undergoes covalent addition with mutant p53 residues Cys124, Cys229, and Cys277, which are the most exposed and reactive residues.

**Figure 14 pharmaceuticals-16-00024-f014:**
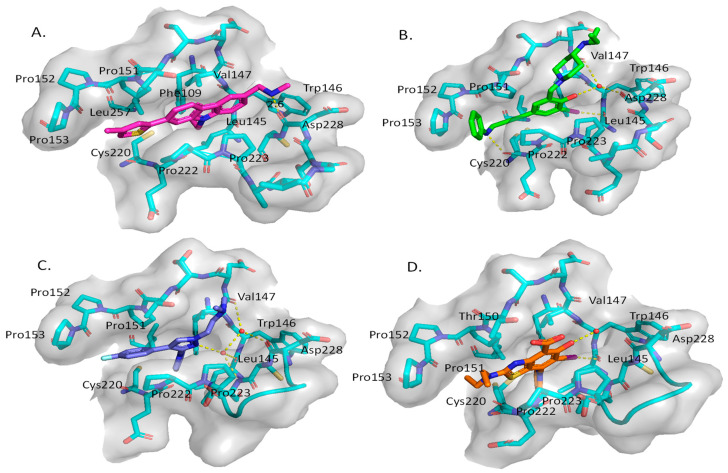
Co-crystal structure of binders of Y220C: (**A**) PK9328; (**B**) PK5196; (**C**) PK7242; (**D**) and MB710.

**Figure 15 pharmaceuticals-16-00024-f015:**
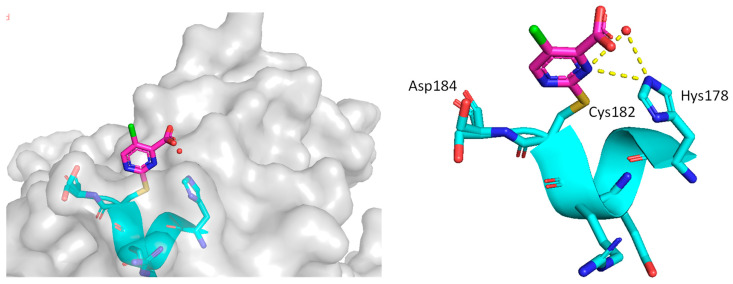
Co-crystal structure of PK11000 covalently bound to Cys182 of a p53 mutant.

**Table 1 pharmaceuticals-16-00024-t001:** Available data of MDM2 inhibitors.

Inhibitor	Target Binding	Cell Potency	In Vivo	Development Stage	PDB	Reference
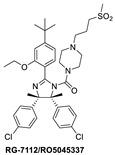	MDM2 HTRF IC_50_ = 18 nM	wt-p53 cells (HCT-116, SJSA-1, RKO) Average MTT IC_50_ = 400 nM	SJSA-1, Tumor regression with over 100 mg/kg PO, QD	Clinical trials	4IPF	[36]
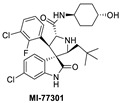	MDM2 K_i_ = 0.88 nM	wt-p53 cells SJSA-1 IC_50_ = 92 nM RS4;11 IC_50_ = 89 nM HCT-116 IC_50_ = 200 nM	SJSA-1 and RS4;11 100% Tumor regression with 100 mg/kg PO, QD	Clinical trials	5TRF	[37]
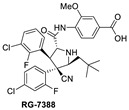	MDM2 HTRF IC_50_ = 6 nM	wt-p53 cells (HCT-116, SJSA-1, RKO) Average MTT IC_50_ = 30 nM	SJSA-1 Tumor regression with 50 mg/kg PO, QD	Clinical trials	Analog co-crystal 4JSC	[38]
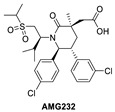	MDM2 HTRF IC_50_ = 0.6 nM	wt-p53 cells SJSA-1 EdU assay IC_50_ = 9.1 nM	SJSA-1 Complete Tumor regression with 60 mg/kg PO, QD	Clinical trials	Analog co-crystal 4OAS	[39,40]
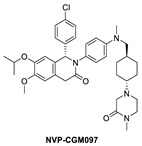	hMDM2 TR-FRET IC_50_ = 1.7 nM hMDM4 TR-FRET IC_50_ = 2000 nM	wt-p53 cells SJSA-1 IC_50_ = 353 nM HCT116 IC_50_ = 454 nM	SJSA-1 Complete Tumor regression with 70 mg/kg PO, 3QW >85% regression 30 mg/kg PO, QD	Clinical trials	4ZYF	[41]
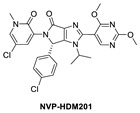	Not disclosed	Not disclosed	10-fold better than NVP-CGM097	Clinical trials	5OC8	[42]
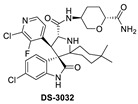	Not disclosed	wt-p53 cells SK-N-SH IC_50_ = 22 nM SH-SY5Y IC_50_ = 18 nM IMR32 IC_50_ = 53 nM IMR5 IC_50_ = 26 nM LAN5 IC_50_ = 44 nM	Wt-TP53 neuroblastoma TGI and prolonged survival	Clinical trials	None	[43]
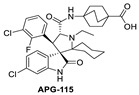	MDM2 IC_50_ = 0.0048 µM	wt-p53 cells SJSA-1 IC_50_ = 60 nM HCT116 IC_50_ = 104 nM	SJSA-1 Complete Tumor regression with over 100 mg/kg PO, QD	Clinical trials	None	[44]
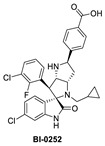	MDM2 IC_50_ = 4 nM	wt-p53 cells SJSA-1 IC_50_ = 471 nM	SJSA-1 Tumor regression with 25 mg/kg PO, QD or 100 mg/kg PO, single dose	BI-907828 in clinical trials	5LAZ	[45]
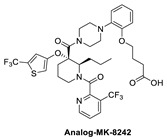	MDM2 IC_50_ = 7 nM	wt-p53 cells SJSA-1 IC_50_ = 80 nM	SJSA-1 TGI = 92% with 50 mg/kg PO, QD 3 days for 4 weeks, or Tumor regression = 60% with 200 mg/kg PO, QD 24 days	NO	5HMH	[46]
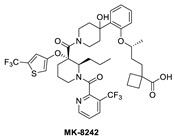	MDM2 K_d_ = 3.8 nM	wt-p53 cells median IC_50_ = 70 nM	Extended survival and tumor regression in various in vivo models	Clinical trials	None	[47,48]
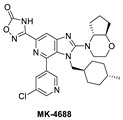	MDM2 TR-FRET IC_50_ = 0.65 nM	wt-p53 cells HCT116 IC_50_ = 122 nM	SJSA-1 Tumor regression with 100 mg/kg PO, QD 7 days	Preclinical	7NA2	[49]

**Table 2 pharmaceuticals-16-00024-t002:** PROTAC–MDM2 degraders.

PROTAC–MDM2 Degrader	Degradation	Cell Potency	In Vivo	Development Stage	Reference
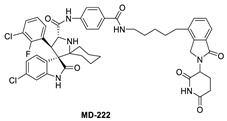	DC_50_ < 1 nM (DC_60_ = 1 nM)	RS4;11 IC_50_ = 2.8 nM	NA	Discovery	[82]
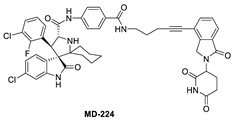	DC_50_ < 1 nM (DC_80_ = 1 nM)	RS4;11 IC_50_ = 1.5 nM	RS4;11 complete tumor regression with 25 mg/kg qd (1–5) or 50 mg/kg qd (1, 3, 5)	Discovery	[82]
(Undisclosed structure) AA-265	DC_50_ < 1 nM	RS4;11 IC_50_ = 0.72 nM	Unpublished	Clinical candidate	
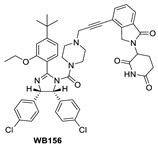	DC_50_ = 22.6 nM	RS4;11 IC_50_ = 3.2 nM	NA	Discovery	[84]
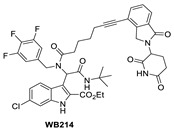	MDM2 DC_50_ = 4.1 nM p53 DC_50_ = 29 nM	RS4;11 IC_50_ = 1.2 nM	NA	Discovery	[85]
(Undisclosed structure) KT-253	DC_50_ = 0.4 nM	RS4;11 IC_50_ = 0.3 nM	RS4;11 complete tumor regression with single dose at 1 mg/kg, or 3 mg/kg, or 10 mg/kg	Clinical candidate	[87]
(Undisclosed structure) XY-27	Undisclosed	Undisclosed	No data	Discovery	[86]

**Table 3 pharmaceuticals-16-00024-t003:** MDMX inhibitors.

MDMX Inhibitors	Target Binding	Cell Potency	In Vivo	Development Stage	PDB	Reference
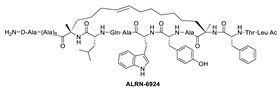	MDM2 IC_50_ = 7.7 nM MDMX IC_50_ = 24.7 nM	MOLM13 (24 h) IC_50_ = 1.4 µM MOLM14 (24 h) IC_50_ = 1.4 µM ML2 (24 h) IC_50_ = 7.9 µM OCI/AML5 (24 h) IC_50_ = >10 µM OCI/AML5 (24 h) IC_50_ = 3.6 µM	Improved survival in AML Xenograft models	Clinical trials	NA	[98]
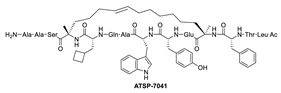	MDM2 K_i_ = 0.9 nM MDMX K_i_ = 6.8 nM	SJSA-1 IC_50_ = 50 nM (1% FBS) IC_50_ = 600 nM (10% FBS)	SJSA-1 (iv) 15 mg/kg or 30 mg/kg achieved TGI = 61% MCF-7 (iv) 20 mg/kg TGI = 63% and 30 mg/kg TGI = 87%	NO	4N5T	[97]
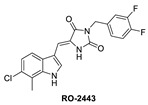	MDM2 IC_50_ = 33 nM MDMX IC_50_ = 41 nM	Poor solubility	NA	NO	3U15	[8]
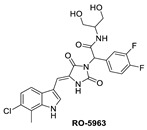	MDM2 IC_50_ = 17 nM MDMX IC_50_ = 24 nM	Active in WT-p53 cells MCF-7, HCT-116, RKO; Inactive in MT-p53 cells SW480, MDA-MB435	NA	NO	NA	[8]
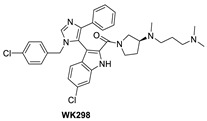	MDM2 FP IC_50_ = 109 nM MDMX FP IC_50_ = 19,700 nM	NA	NA	NO	3LBJ	[101]
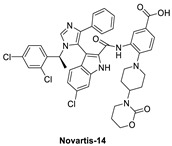	MDM2 TR-FRET IC_50_ < 0.1 nM MDMX TR-FRET IC_50_ = 17 nM	NA	NA	NO	6Q9S	[99]

**Table 4 pharmaceuticals-16-00024-t004:** Mutant p53 activators.

Mutant p53 Reactivators	Target Binding	Cell Potency	In Vivo	Development Stage	PDB	Reference
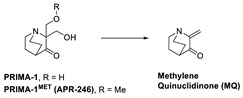	Covalent binding to Cys124, Cys229, Cys277	Saos-2 (His-273 mutant p53) PRIMA-1 IC_50_ = 14 µM, APR246 IC_50_ = 9 µM; H1299 (His-175 mutant p53) PRIMA-1 IC_50_ = 24 µM, APR246 IC_50_ = 19 µM	PRIMA-1: Saos-2-His-273 xenograft IV, BID, 3 days at 20 mg/kg or 100 mg/kg achieve 90% and 98% TGI, respectively	APR-246 in clinical trials	NA	[125,128]
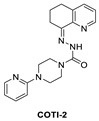	NA	nanomolar (nM) inhibition of proliferation after 72 h treatment of a diverse group of cancer cell lines, regardless of tissue origin and genetic background	Significant TGI in SHP-77 (Human SCLC) xenograft at 3 mg/kg IV; and in OVCAR3 (Human Ovarian Carcinoma) xenografts 20 mg/kg IV or 75 mg/kg PO	Clinical trials	NA	[134]
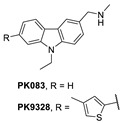	Mutant p53 Y220C PK083 K_d_ = 125 µM PK9328 K_d_ = 1.7 µM	PK9328 100% viability loss at 7.5 µM in HUH7 cells ~50% viability loss at 7.5 µM in HUH7 p53 KO cells	NO	Discovery	PK083 PDB: 2VUK PK9328 PDB: 6GGF	[137,139]
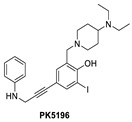	Mutant p53 Y220C K_d_ = 9.7 µM	50 µM induced apoptosis onset in NUGC-3 (Y220C mutant p53) and No apoptosis in NUGC-4 (wt-p53)	NO	Discovery	4AGQ	[140]
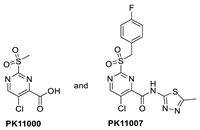	Covalent binding Cys182, Cys277	PK11007: Viability reduction at 15 to 30 µM in mutant p53 cell lines MKN1(V143A), HUH-1(Y220C), NUGC-3(Y220C), and SW480(R273H/P309S) and wt-p53 HUH-6, NUGC-4, and WI38 cells were less sensitive; reducing viability between 60 and 120 µM also has p53-independent activity	NO	Discovery	PK11000 PDB: 5LAP	[143]
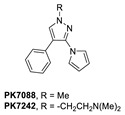	Mutant p53 Y220C K_d_ = 140 µM	PK7088 at 200 µM induces apoptosis in NUGC-3, HUH-7 (Y220C mutant p53 cells), and minimal effect in NUGC-4, HUH-6 (wt-p53)	NO	Discovery	PK7242 PDB: 3ZME	[141]
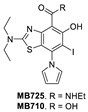	MB725, K_d_ = not soluble MB710, K_d_ = 4 µM	IC_50_s Y220C mutant p53 NUGC3 MB725 = 10 µM, MB710 = 90 µM; BXPC3 MB725 = 18 µM, MB710 = ND; HUH7 MB725 = 10 µM, MB710 = ND; and in wt-p53 NUGC4, WI38 MB725, M710 >120 µM; HUH6 MB725, MB710 = ND; and in R273C mutant p53 SW1088 MB725, MB710 >120 µM	NO	Discovery	MB710 PDB: 5O1I	[142]
PC14586 (Undisclosed structure)	Y220C binder	Lymphoma and sarcoma cell lines generated from HUPKI-Y220C mouse tumors were sensitive to Y220C p53 reactivator PC14586 IC_50_ ~192–722 nM	HUPKI Y220C mutant sarcoma cell xenograft 100 mg/kg QD antitumor activity with two cures and synergy with 50 mg/kg PC14586 QD + 200 µg anti-PD-1 Q3D	Clinical trials	NA	[144]

**Table 5 pharmaceuticals-16-00024-t005:** Clinical trials of wild-type and mutant p53 activators.

Drug	Target	Development Phase	Combination	Disease	NCT Number	Status
RG7112	MDM2–p53 inhibitor	Phase I		Myelogenous Leukemia, Chronic Neoplasms, Myelogenous Leukemia, Acute	NCT01677780	Completed
		Phase I		Neoplasms	NCT01164033	Completed
		Phase I	Doxorubicin	Sarcoma	NCT01605526	Completed
		Phase I		Sarcoma	NCT01143740	Completed
		Phase I		Hematologic Neoplasms	NCT00623870	Completed
		Phase I		Neoplasms	NCT00559533	Completed
		Phase I	Cytarabine	Acute Myelogenous Leukemia	NCT01635296	
MI773	MDM2–p53 inhibitor	Phase I		Neoplasm Malignant	NCT01636479	Completed
		Phase I	Pimasertib	Neoplasm Malignant	NCT01985191	Completed
RG7388 (Idasanutlin)	MDM2–p53 inhibitor	Phase I	Pegasys	Polycythemia Vera, Essential Thrombocythemia	NCT02407080	Completed
		Phase III	Cytarabine Placebo	Leukemia, Myeloid, Acute	NCT02545283	Terminated
		Phase II		Polycythemia Vera	NCT03287245	Terminated
		Phase I	Placebo [13C]-radiolabeled Idasanutlin [14C]-radiolabeled Idasanutlin	Solid Tumors	NCT02828930	Completed
		Phase I		Solid Tumors	NCT03362723	Completed
		Phase I, II	Cytarabine Daunorubicin Allogeneic Hematopoietic Stem Cell Transplant (Allo-HSCT)	Acute Myeloid Leukemia	NCT03850535	Terminated
		Phase I, II	Venetoclax Cyclophosphamide Topotecan Fludarabine Cytarabine Intrathecal Chemotherapy	Acute Myeloid Leukemia (AML), Acute Lymphoblastic Leukemia (ALL), Neuroblastoma, Solid Tumors	NCT04029688	Recruiting
		Phase I, II	Dexamethasone Ixazomib Citrate	Loss of Chromosome 17p, Recurrent Plasma Cell Myeloma	NCT02633059	Active, not recruiting
		Phase I, II	Obinutuzumab Venetoclax Rituximab	Follicular Lymphoma, Lymphoma, Large B Cell, Diffuse	NCT03135262	Terminated
		Phase I, II	Obinutuzumab Rituximab	Non-Hodgkin Lymphoma	NCT02624986	Terminated
		Phase I	Cobimetinib Venetoclax	Leukemia, Myeloid, Acute	NCT02670044	Completed
		Phase I, II	Atezolizumab Cobimetinib	Stage III, IIIA, IIIB, IIIC, IV Breast Cancer, Estrogen Receptor-positive, HER2/Neu Negative	NCT03566485	Terminated
		Phase II	Entrectinib Alectinib Atezolizumab Ipatasertib Trastuzumab emtansine Inavolisib Belvarafenib Pralsetinib	Solid Tumors	NCT04589845	Recruiting
		Phase I, II	Regorafenib Atezolizumab Imprime PGG Bevacizumab Isatuximab Selicrelumab AB928 Genetic: LOAd703	Colorectal Cancer	NCT03555149	Active, not recruiting
		Phase I, II	APG101 Alectinib Atezolizumab Vismodegib Temsirolimus Palbociclib	Glioblastoma, Adult	NCT03158389	Recruiting
AMG232	MDM2–p53 inhibitor	Phase I		Advanced Malignancy, Advanced Solid Tumors, Glioblastoma, Multiple Myeloma	NCT01723020	Completed
		Phase I	Trametinib	Advanced Malignancy, Acute Myeloid Leukemia	NCT02016729	Completed
		Phase I	Trametinib Dabrafenib	Advanced Malignancy, Advanced Solid Tumors, Melanoma	NCT02110355	Completed
		Phase I	Navtemadlin Radiation: Radiation Therapy	Glioblastoma, Gliosarcoma, MGMT-Unmethylated Glioblastoma, Recurrent Glioblastoma	NCT03107780	Suspended
		Phase I	Carfilzomib Dexamethasone Dexamethasone Sodium Phosphate Lenalidomide Navtemadlin	Plasmacytoma, Recurrent Plasma Cell Myeloma, Refractory Plasma Cell Myeloma	NCT03031730	Recruiting
		Phase I	Decitabine Navtemadlin	Acute Myeloid Leukemia, Recurrent Acute Myeloid Leukemia, Refractory Acute Myeloid Leukemia, Secondary Acute Myeloid Leukemia	NCT03041688	Suspended
		Phase I	Navtemadlin Radiation: Radiation Therapy	Resectable Soft Tissue Sarcoma, Soft Tissue Sarcoma	NCT03217266	Active, not recruiting
		Phase I	Cytarabine Idarubicin Hydrochloride Navtemadlin	Acute Myeloid Leukemia, Acute Myeloid Leukemia Arising From Previous Myelodysplastic Syndrome	NCT04190550	Recruiting
NVPCGM097	MDM2–p53 inhibitor	Phase I		Solid Tumor With p53 Wild-Type Status	NCT01760525	Completed
NVPHDM201	MDM2–p53 inhibitor	Phase I, II	Pazopanib	Advanced Soft-Tissue Sarcoma, Metastatic Soft-Tissue Sarcoma	NCT05180695	Recruiting
		Phase I	Midostaurin	AML, Adult	NCT04496999	Recruiting
		Phase I	Trametinib	Colorectal Cancer, Advanced Cancer, Metastatic Cancer	NCT03714958	Recruiting
		Phase I	LEE011	Liposarcoma	NCT02343172	Completed
		Phase I	Ancillary treatment	Advanced Solid and Hematological WT-TP53 Tumors	NCT02143635	Completed
		Phase I	MBG453 Venetoclax	Acute Myeloid Leukemia (AML), High-Risk Myelodysplastic Syndrome (MDS)	NCT03940352	Recruiting
		Phase I, II	cytarabine anthracycline midostaurin liposomal cytarabine/daunorubicin posaconazole midazolam	Leukemia, Myeloid, Acute	NCT03760445	Withdrawn
		Phase I	LXS196	Uveal Melanoma	NCT02601378	Terminated
		Phase I, II	Siremadlin	Acute Myeloid Leukemia, Allogeneic Stem Cell Transplantation	NCT05447663	Not yet recruiting
		Phase I, II	siremadlin venetoclax azacitidine	Acute Myeloid Leukemia	NCT05155709	Recruiting
		Phase I, II	Ruxolitinib Siremadlin Crizanlizumab Sabatolimab LTT462 NIS793	Myelofibrosis	NCT04097821	Active, not recruiting
		Phase II	Ribociclib Cabozantinib Alectinib Regorafenib Trametinib Dabrafenib	Malignant Solid Tumor	NCT04116541	Recruiting
		Phase I	PDR001 LCL161 Everolimus Panobinostat QBM076	Colorectal Cancer, Non-Small Cell Lung Carcinoma (Adenocarcinoma), Triple Negative Breast Cancer, Renal Cell Carcinoma	NCT02890069	Completed
DS3032 (Milademetan)	MDM2–p53 inhibitor	Phase I		Myeloma	NCT02579824	Terminated
		Phase I, II	Cytarabine Venetoclax	Acute Myeloid Leukemia, Recurrent Acute Myeloid Leukemia, Refractory Acute Myeloid Leukemia	NCT03634228	Completed
		Phase I		Acute Myeloid Leukemia	NCT03671564	Completed
		Phase I	AZA	Acute Myelogenous Leukemia, Myelodysplastic Syndrome	NCT02319369	Terminated
		Phase I	Quizartinib	Acute Myeloid Leukemia	NCT03552029	Terminated
		Early Phase I	Itraconazole Posaconazole	Pharmacokinetics	NCT03614455	Completed
		Phase I		Advanced Solid Tumor, Lymphoma	NCT01877382	Completed
		Early Phase I		Food Effects on Pharmacokinetics	NCT03647202	Completed
APG115 (Alrizomadlin)	MDM2–p53 inhibitor	Phase I		Patients With Advanced Solid Tumor or Lymphoma	NCT02935907	Completed
		Phase II	APG-2575	T-Prolymphocytic Leukemia	NCT04496349	Recruiting
		Phase I, II	Toripalimab	Liposarcoma, Advanced Solid Tumor	NCT04785196	Recruiting
		Phase I, II	5-azacitidine	Acute Myeloid Leukemia, Chronic (AML), Myelomonocytic Leukemia (CMML), Myelodysplastic Syndromes, High-Risk Myelodysplastic Syndrome, MDS	NCT04358393	Recruiting
		Phase I, II	Pembrolizumab	Unresectable or Metastatic Melanoma or Advanced Solid Tumors, Melanoma, Uveal Melanoma, P53 Mutation, MDM2 Gene Mutation, MPNST, Cutaneous Melanoma, Mucosal Melanoma, Malignant Peripheral Nerve Sheath Tumors	NCT03611868	Recruiting
		Phase I, II	Carboplatin	Malignant Salivary Gland Cancer, Salivary Gland Cancer	NCT03781986	Recruiting
		Phase I	Azacitidine Cytarabine	Acute Myeloid Leukemia (AML), Myelodysplastic Syndromes (MDS)	NCT04275518	Recruiting
BI-907828	MDM2–p53 inhibitor	Phase I	Rifampicin	Solid Tumors	NCT05372367	Recruiting
MK-8242	MDM2–p53 inhibitor	Phase I		Acute Myelogenous Leukemia (AML)	NCT01451437	Terminated
		Phase I		Solid Tumors	NCT01463696	Terminated
ALRN-6924	MDM2/MDMX inhibitor	Phase I	cytarabine	Acute Myeloid Leukemia, Myelodysplastic Syndromes	NCT02909972	Completed
		Phase I	Paclitaxel	Advanced Malignant Solid Neoplasm, Anatomic Stage III Breast Cancer AJCC v8, Anatomic Stage IIIA Breast Cancer AJCC v8, Anatomic Stage IIIB Breast Cancer AJCC v8, Anatomic Stage IIIC Breast Cancer AJCC v8, Estrogen Receptor Positive, HER2/Neu Negative, Metastatic Malignant Solid Neoplasm, Prognostic Stage III Breast Cancer AJCC v8, Prognostic Stage IIIA Breast Cancer AJCC v8, and 5 more	NCT03725436	Recruiting
		Phase I, II		Solid Tumor, Lymphoma, Peripheral T-Cell Lymphoma	NCT02264613	Completed
		Phase I	Cytarabine	Leukemia, Brain Tumor, Solid Tumor, Lymphoma	NCT03654716	Recruiting
		Phase I	Carboplatin, Pemetrexed, Placebo, Topotecan	Non-Small Cell Lung Cancer, Small-Cell Lung Cancer	NCT04022876	Active, not recruiting
APR-246 (eprenetapopt)	Mutant p53 activator	Phase I, II		Oesophageal Carcinoma	NCT02999893	Terminated
		Phase I	Venetoclax, Azacitidine	Myeloid Malignancy	NCT04214860	Completed
		Phase II		Acute Myeloid Leukemia or Myelodysplastic Syndromes	NCT03931291	Completed
		Phase I, II	Azacitidine	Myelodysplastic Syndrome With Gene Mutation, Acute Myeloid Leukemia With Gene Mutations, Myeloproliferative Neoplasm, Chronic Myelomonocytic Leukemia	NCT03588078	Unknown
		Phase I, II	Pembrolizumab	Bladder Cancer, Gastric Cancer, Non-Small Cell Lung Cancer, NSCLC, Urothelial Carcinoma, Advanced Solid Tumor	NCT04383938	Completed
		Phase I, II	+ Acalabrutinib in CLL; + Venetoclax and Rituximab in CLL; (Acalabrutinib, OR, (Venetoclax +Rituximab)), in CLL and/or MCL and/or RT; Venetoclax + Rituximab in RT	Non-Hodgkin Lymphoma, Chronic Lymphocytic Leukemia, Mantle Cell Lymphoma	NCT04419389	Suspended
		Phase III	Azacitidine	MDS	NCT03745716	Completed
		Phase II	Pegylated Liposomal Doxorubicin Hydrochloride (PLD)	High-Grade Serous Ovarian Cancer	NCT03268382	Completed
		Phase I, II	Azacitidine	Myelodysplastic Syndrome, Acute Myeloid Leukemia, Myeloproliferative Neoplasm, Chronic Myelomonocytic Leukemia	NCT03072043	Completed
		Phase I		Hematologic Neoplasms, Prostatic Neoplasms	NCT00900614	Completed
		Phase I, II	Carboplatin and Pegylated Liposomal Doxorubicin Hydrochloride (PLD)	Platinum Sensitive Recurrent High-Grade Serous Ovarian Cancer With Mutated p53	NCT02098343	Completed
		Phase I, II	Dabrafenib	Melanoma	NCT03391050	Terminated
		Phase II	Venetoclax	Recurrent Mantle Cell Lymphoma; Refractory Mantle Cell Lymphoma	NCT04990778	Withdrawn
COTI-2	Mutant p53 activator	Phase I	Cisplatin	Ovarian Cancer, Fallopian Tube Cancer, Endometrial Cancer, Cervical Cancer, Peritoneal Cancer, Head and Neck Cancer, HNSCC, Colorectal Cancer, Lung Cancer, Pancreatic Cancer	NCT02433626	Unknown
PC14586	Y220C Mutant p53 activator	Phase I, II		Advanced Solid Tumor, Advanced Malignant Neoplasm, Metastatic Cancer, Metastatic Solid Tumor	NCT04585750	Recruiting
		Phase I		Healthy Volunteers	NCT05249348	Recruiting
		Phase I		Healthy Male Volunteers	NCT05523687	Recruiting

## Data Availability

Not applicable.

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
