# Peer review of "Therapeutic Strategies to Activate p53"

_pharmaceuticals, 2022, doi:10.3390/ph16010024_

Round 1

Reviewer 1 Report

The manuscript sounds quite interesting. The review is very well-written foccusing on recent therapeutic advances in targeting the major negative regulation MDM2-MDMX-p53 axis and activation of one specific p53 Y220C mutant. The manuscript provides important data at molecular level targeting not only the protein-protein interaction between MDM2 and p53, but  also the MDM2-p53 protein-protein interactions disruption, pointing out MDMDX inhibitors. Mutant p53 Y220C binders and activators are also explored in this review and the authors listed clinical trials in progress providing material to new contributions in cancer field.

Additional comments:

p.1 line 44:  wild type (wt) - the initials should be firstly defined.
fig 1: nutrient instead of nutreint (typo error)

Author Response

Dear Reviewer:

We thank you for your speedy review and positive decision on our manuscript.  Below are our point-to-point responses of your suggestions. These corrections are highlighted for easy viewing in the manuscript.

Point 1: p.1 line 44: wild type (wt) - the initials should be firstly defined.

Response 1: The initials have been defined “wild-type (wt)” in the manuscript

Point 2: fig 1: nutrient instead of nutreint (typo error)

Response 2: “nutreint” has been changed to “nutrient” in figure 1.

Reviewer 2 Report

The manuscript by Aguilar and Wang describes recent advances in the development of anticancer drugs that lead activation of the tumor suppressor gene product p53. Overall, it is well organized, but there are some concerns. Therefore, it can be published in Pharmaceuticals after minor revisions. Details are given below.

Minor points:

1.       Line 168-169: Shouldn't it be "residues of MDM2 necessary for binding to p53"?

2.       Lines 247-249: Is the sentence an appropriate way to describe it?

3.       Line 420: A brief description of GSPT-1 should be added.

4.       Line 651,"the mechanism of action (MOA)": MOA is already described in line 439.

5.       The size of the text used in the figures should be increased.

6.       Figure 1: A detailed explanation should be added to the legend. The text and legend contain no descriptions of certain proteins in the figure. 

Author Response

Dear Reviewer:

We thank you for your speedy review, constructive suggestions and positive decision on our manuscript.  Below are our point-to-point responses of your suggestions.  These corrections are highlighted for easy viewing in the manuscript. 

Point 1: Line 168-169: Shouldn't it be "residues of MDM2 necessary for binding to p53"?

Response 1: Thank you for your suggestion but your statement is inverted. To clarify, the sentence has been reworded as follows. “This co-crystal structure indicates that the isopropyl, para-chlorophenyl, and the meta-chlorophenyl substituents occupy the three key binding pockets on MDM2, mimicking the interactions of the p53 residues Phe19, Trp23 and Leu26, respectively.”

Point 2: Lines 247-249: Is the sentence an appropriate way to describe it?

Response 2: The sentence has been reworded as follows. “Unique in DS-3032 is the 2-chloro-3-fluoro-pyridine substituent at the C3 of its pyrrolidine core. All other reported spiro-oxindoles in clinical trials contain a 2-fluoro-3-chlorophenyl substituent at this position (Figure 2).”

Point 3: Line 420: A brief description of GSPT-1 should be added.

Response 3: A brief description has been added to the sentence as follows. “Further research revealed that this structural modification converted the chimera into a molecular glue that degraded the G1 to S phase transition 1 (GSPT-1) protein, a translation termination factor, also known as eRF3a, that mediates stop codon recognition and nascent protein release from the ribosome through the interaction with release factor , eRF1 (Figure 8).”

Point 4: Line 651,"the mechanism of action (MOA)": MOA is already described in line 439.

Response 4: The statement “the mechanism of action (MOA)” has been replaced with “the MOA” as requested.

Point 5: The size of the text used in the figures should be increased.

Response 5: The size of the text in the figure 1 to figure11 have been increased.

Point 6: Figure 1: A detailed explanation should be added to the legend. The text and legend contain no descriptions of certain proteins in the figure.

Response 6: Figure 1 was edited and a detailed explanation was added to the legend.